# Technological Advances and Trends in Modern High-Rise Buildings

**Jerzy Szolomicki [1],\* and Hanna Golasz-Szolomicka [2]**

[1]    Faculty of Civil Engineering, Wroclaw University of Science and Technology, 50-370 Wroclaw, Poland
[2]    Faculty of Architecture, Wroclaw University of Science and Technology, 50-370 Wroclaw, Poland
\*    Correspondence: Jerzy.Szolomicki@pwr.edu.pl; Tel.: +48-505-995-008

**Abstract:** The purpose of this paper is to provide structural and architectural technological solutions applied in the construction of high-rise buildings, and present the possibilities of technological evolution in this field. Tall buildings always have relied on technological innovations in engineering and scientific progress. New technological developments have been continuously taking place in the world. It is closely linked to the search for efficient construction materials that enable buildings to be constructed higher, faster and safer. This paper presents a survey of the main technological advancements on the example of selected tall buildings erected in the last decade, with an emphasis on geometrical form, the structural system, sophisticated damping systems, sustainability, etc. The famous architectural studios (e.g., for Skidmore, Owings and Merill, Nikhen Sekkei, RMJM, Atkins and WOHA) that specialize, among others, in the designing of skyscrapers have played a major role in the development of technological ideas and architectural forms for such extraordinary engineering structures. Among their completed projects, there are examples of high-rise buildings that set a precedent for future development.

**Keywords:** high-rise buildings; development; geometrical forms; structural system; advanced materials; damping systems; sustainability

## 1. Introduction

High-rise buildings play an increasingly important role in contemporary architecture. Their raising is a necessity for the process of population growth and its concentration in cities, as well as for the high demand for areas in city centers [1]. It can be observed the dynamic development of their construction in terms of both quantity and quality [2]. There are plans to build 219 high-rise buildings worldwide in 2019. According to the Global Tall Buildings Database of the CTBUH (Council on Tall Buildings and Urban Habitat) until now were erected 1647 buildings taller than 200 m. The high-rise building construction is characterized by high demand of construction technology and complex engineering works [3].

In contemporary architecture, designers go beyond the framework of standard codified construction assumptions in order to provide additional and unusual aesthetic experiences [4]. Geometric shapes, impressive in terms of body and scale, are used for this purpose, as well as the newest material technologies, thanks to which skyscrapers can be classified as eco-buildings.

The change in the approach in building design in the last two decades is reflected in the models for shaping a sustainable, energy-saving environment, which are specified in the context of comparable methods for assessing buildings with various criteria (quality assessment tools, including Leed). These changes are evidenced by many documents, including the Aalborg Charter [5], the European Charter for Solar Energy in Architecture and Urban Planning [6], and the White Book of the Architects' Council

of Europe [7]. Energy-efficient architecture is promoted by such architects as Norman Foster [8], Renzo Piano [9], Thomas Herzog [10] and Gilles Perraudin [11].

The main trend among new high rise buildings is the striving to achieve zero energy, which is associated with Leed certification [12]. Obtainment of Leed v4 certification at the Platinum level means the highest green building standard in the world. Bryant Park (New York, NY, USA) became the first high-rise building in the world to attain this certificate. Other buildings to achieve the Leed v4 certificate include, among others, Shanghai Tower (Shanghai, China), Taipei 101 (Taipei, Taiwan) and Hearst Tower (New York, NY, USA).

One of the pro-ecological ideas is the design of bioclimatic skyscrapers, in which users' comfort is increased by greenery inside the buildings through the use of public terraces or multi-level atrias (Oasia Hotel, Singapore).

The problem of high building design particularly concerns problems related to the limitation of horizontal displacements of the building and ensuring its spatial rigidity, proper foundation and resistance to dynamic wind action and seismic effects. The key design challenge associated with acting loads is the appropriate selection of the structural system, while at the same time optimizing its geometrical dimensions. The existing construction solutions mainly differ in their way of transmitting horizontal forces from the wind and seismic impacts on the foundations. A sophisticated construction system allows building in seismic areas with strong wind (Tokyo Sky-tree, Tokyo) and artificially created land (United Tower, Manama; Marina Bay Sands complex, Singapore).

The paper presents the architectural and constructional characteristics of selected modern high-rise buildings, which follow the trend constitutes an architectural paradigm that focuses on sustainable design.

## 2. Methods

The high-rise buildings implemented today are astonishing in terms of the multitude of their architectural and constructional solutions, as well as their technology. Conducting a comprehensive analysis of technological innovations used in these buildings, due to the size of the issue, requires a special approach. Therefore, a methodology was developed that includes the following elements:

- Gathering information on innovative technologies of modern high-rise buildings and completing photos and videos documenting their erection,
- Interpretation of collected information on the basis of literature, generally addressing the problem of advanced technologies used in completed buildings,
- Conducting, according to a structured diagram, architectural and construction analysis of selected high-rise buildings, in which the applied technologies were significantly more advanced than those of previous projects.

Such a system facilitated the conducting of a structured analysis, with particular emphasis being put on the building's body, construction system, vibration damping system, ultra-strong concrete and steel, low-emission glass, double or triple skin facades and elements decisive for energy saving of the building.

## 3. Technological Innovations in High-Rise Buildings

### 3.1. New Design Trends in Geometrical Forms

High-rise buildings were often designed in the form of rectangular blocks with glass façades. Such buildings, although practical and aesthetic, are somewhat monotonous. Contemporary architecture is trying to face this monotony. Apart from the mass execution of rational high-rise buildings, the appearance of another trend has been noted. This is the phenomenon of erecting "iconic" buildings, which are distinguished by their shape and scale. Based on the information gathered in the CTBUH database and also looking at new created high-rise buildings, it is reasonable to believe that the next

generation of tall buildings will be more towards aerodynamic and curvilinear shapes and forms. Analysis of wind action on tall buildings shows the importance of the effect of form and geometry of high-rise buildings. For instance, in Taipei 101, corner modifications provide 25% reduction in the base moment when compared to the original square section.

Geometric solids (such as polyhedra, cones, cylinders, spheres, ellipsoids and toruses) and curved surfaces appear as the components of each modern skyscraper [13].

Analyzing the form of a building can identify the specific types of basic solids or surfaces used in all or part of the building. In addition to creating the composition of a building, different kinds of distortion of the solids or surfaces are used. Generally, spatial forms can be geometrically divided into polyhedra, solids of revolution and surfaces [14]. Polyhedra are solids limited by a closed surface and constructed with a finite number of flat polygons. They are divided into: Prismatoids (pyramids, prisms, anti-prisms and octahedrons), polyhedra (platonic solids), semiregular polyhedra (the Archimedean solids) and other polyhedra compounds (Catalan, Johnson and toroidal).

The second group of geometric forms is the solids of revolution, which are limited by a closed surface of revolution or toroidal based on a circle, ellipse or other closed figure. In them are a sphere, ellipsoid or torusoid of revolution with a normal section in the shape of a circle or ellipse.

The last group of spatial forms is surfaces, which include: Ruled surfaces (Catalana, conical and cylindrical), curved surfaces of a constant generatix (rotary, torusoidal and translational) and curved surfaces of a varying generatix (wedge, parabolic-elliptic and minimum). From an architectural point of view, modern skyscrapers can be categorized into the following groups: Extruder, rotor, twister, tordos and also free form.

### 3.1.1. Extruder

Buildings of this type have the same cross section for their full height. An example can be a rectangular or cylindrical solid (Figure 1a). Within this group the following modifications may exist:

- Individual stories are arranged one on the other in a constant slope and the floor plans may have a straight or curved contour ("anglers", Figure 1b);
- Stories are arranged one on the other, sometimes with a different angle of inclination and often in the form of straight segments inclined in different directions that are smoothly connected to the curved segments ("sliders", Figure 1c), which may be narrowing with an increasing height ("tapering sliders", Figure 1d);
- Single buildings of this type can be connected together in groups in order to provide increased rigidity or fire protection exit ("slider assemblies", Figure 1e).

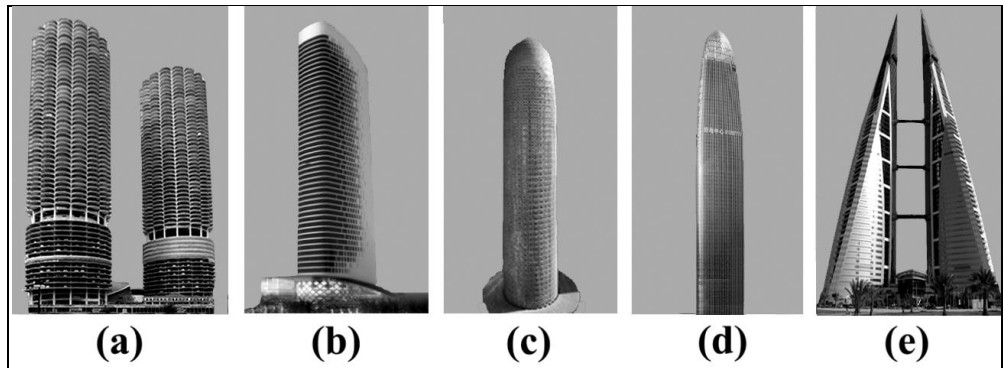

**Figure 1.** Extruder type of high-rise buildings: (**a**) Marina City towers (Chicago, cylindrical central core system), (**b**) Bay Gate (Dubai, wall frame system), (**c**) Doha tower (Doha, tube frame system), (**d**) Greenland Puli Center (Jinan, core and outrigger system) and (**e**) the World Trade Center (Bahrain, shear wall frame system), figure by authors.

### 3.1.2. Rotor

Buildings of this type have the same cross section for their full height. An example can be a rectangular or cylindrical solid (Figure 2a). Within this group the following modifications may exist:

- Individual stories are arranged one on the other in a constant slope and the floor plans may have a straight or curved contour ("anglers", Figure 2b);
- Stories are arranged one on the other, sometimes with a different angle of inclination and often in the form of straight segments inclined in different directions that are smoothly connected to the curved segments ("sliders", Figure 2c), which may be narrowing with an increasing height ("tapering sliders", Figure 2d);
- Single buildings of this type can be connected together in groups in order to provide increased rigidity or fire protection exit ("slider assemblies", Figure 2e).

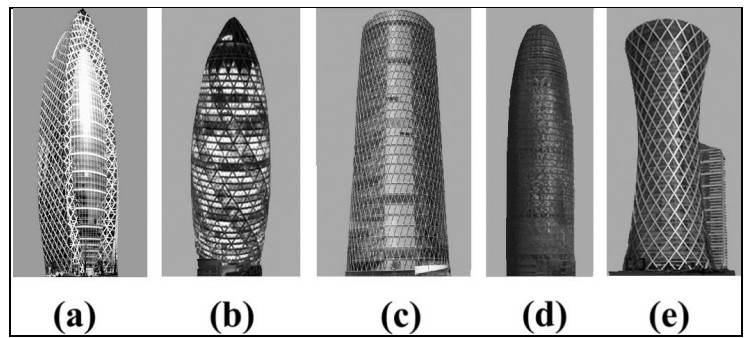

**Figure 2.** Rotor type of high-rise buildings: (**a**) Mode Gakuen Cocoon Tower (Tokyo, tube system with concrete columns), (**b**) Swiss Re (London, diagrid frame tube system), (**c**) Westhafen Tower (Frankfurt, diagrid frame tube system), (**d**) Torre Agbar (Barcelona, diagrid frame tube system) and (**e**) Tornado Tower (Doha, concrete core system with an external tubular steel diagrid), figure by authors.

### 3.1.3. Twister and Tordos

Buildings of this type are in the form of a twisted solid with the "twister" facade repeated on all floors (Figure 3a). Buildings with an orthogonal core and one or two twisted towers belong to the category of "toros" (Figure 3b). The conversion tower axis of the helical form the Revolution Tower (Figure 3c) may be derived from orientation of asymmetric floors not through the center of the circular segment stories, but the center of gravity of the floor. The body of the building belongs to the category of "sliding twister" (Figure 3c), where the floors are moved upward along the 2D or 3D curve and rotation is added to the outer structure. When the 3D curve rotation has the shape of a spiral it belongs to the category of "helical twister" (Figure 3d,e). The intersecting body of the building, in the shape of a twisted spiral, has an internal vertical zone dedicated to the lift shaft.

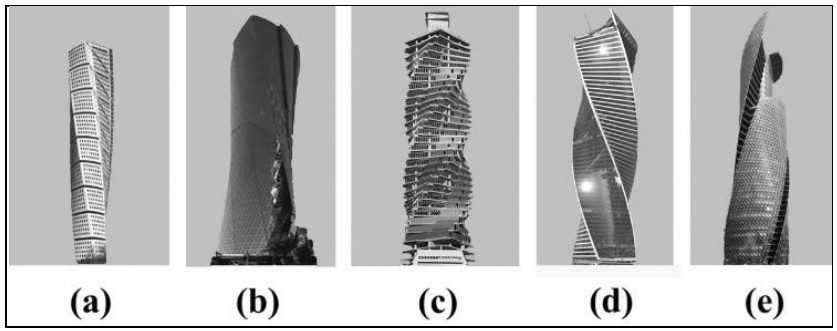

**Figure 3.** Twister and Tordos types of high-rise building: (**a**) Turning Torso (Malmö, mega core system), (**b**) Al Bidda Tower (Doha, wall frame system), (**c**) Revolution Tower (Panama City, core system), (**d**) Evolution Tower (Moscow, core system)) and (**e**) Mode Gakuen Spiral Towers (Nagoya, tube system), figure by authors.

### 3.1.4. Free Form

The free geometry building form is constructed using a combination of geometrically simple objects (lines, surfaces and solids), when the sequence of the architect's actions is not obvious and the form does not fit into any other category. In this category we can distinguish the subcategory "slicer". It includes buildings that have a curved facade with balconies and other extended elements. Figure 4a shows the curved outer surface obtained by the contours of winding balconies around a rectangular solid. Alternatively, the curved segments of the balconies can be repeated on the upper floors with their rotation (Figure 4b). This vertical twisting of the outer surface is formed in a cross-section that is not a straight line but a curve. The receding facade of the building in Figure 4b is decorated with flat elements. The smooth surface of the building in Figure 4c is obtained by a large number of blinds. The verticality of railings is less obvious in high-rise buildings than in low buildings (Figure 4d), where the facade is rather stepped and does not create a smoothed curve. The building is classified in the "sliced twister" category (Figure 4b) when it has repeated vertical floors with horizontal rotation.

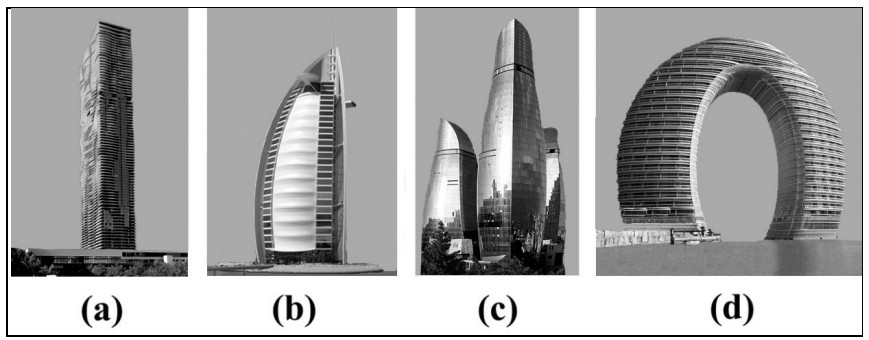

**Figure 4.** Free form type of high-rise buildings: (**a**) Aqua Tower (Chicago, core and outrigger system), (**b**) Burj Al Arab (Dubai, composite frame system with diagonal steel trusses), (**c**) Flame Towers (Baku, frame tube system) and (**d**) Sheraton Huzhou Hot Spring Resort (Huzhou, core system), figure by authors.

High-rise shaping is largely related to the numerical modeling tools that architects have available. Simple modeling procedures enable intuitive shaping of complex geometry, however mathematical analysis is required because the consequences for the structure are considerable.

### 3.2. Innovations in Structural Systems

### 3.2.1. Structural Systems

The relationship between structure and architectural form has reached its peak in present times. Form and structure have become inseparable and complementary [15]. The primary structural skeleton of a high-rise building can be visualized as a vertical cantilever beam with its base fixed in the ground. The structure has to carry vertical gravity loads, the lateral wind and also earthquake loads. The building must therefore have adequate shear and bending resistance and must not lose its vertical load-carrying capability.

Structural systems of tall buildings can be divided into various types due to different criteria (e.g., internal and external). The choice of system and application of constructional material is affected by many factors, in particular:

- The height of the building,
- The ratio of height to width (slenderness),
- The required spatial rigidity for the transfer of lateral forces (wind, seismic),
- The formation of the building's body,
- The conditions of the layout of the lower floor and foundation.

The structural system of high-rise buildings can be divided on the following types [16]: Rigid frame, shear frame (shear trussed frame, shear walled frame), flat plate, mega column (frame, truss), core, mega core, outriggered frame and tube (framed tube, truss tube).

### 3.2.2. Innovative Diagrid System

Currently the diagrid system is one of the most innovative and adaptable approaches to structuring high-rise building (Capital Gate Tower (Abu Dhabi, UAE), Swiss Re (London, UK), Hearst Tower (New York, NY, USA) and CCTV headquarters (Beijing, China)). This kind of structure has evolved from a diagonalized tube. A diagrid is a special form of spatial truss. The difference between a conventional braced-tube structure and the current diagrid structure is that the diagrid system has almost completely eliminated the use of columns [17,18]. This is possible because diagonal elements in the diagrid system can carry gravity loads as well as horizontal loads due to their triangular configuration. The constructional function is realized by transfer lateral loads through the axial action of structural components. The bending stiffness is obtained by a diagonal grid, which also gives the shear stiffness. Adoption of such forms is very beneficial for reasons of dynamic impacts. As the height of a building increases, the lateral strength becomes more important than the load-bearing system that carries gravity loads. Therefore, any modifications to the geometric form of tall buildings generally reduce the adverse effects of the wind, which is an additional reason for the greater creativity of architects.

The diagonal grid module has a trapezoid shape and its height is several floors. Depending on the number of stories, the modules are divided into small (2–4 stories), medium (6–8 stories) and large (over 8 stories). Modules and diagonal angles play a key role in the structural, architectural and aesthetic concept of the design of the building. Due to the form, they may be flat, crystalline or multi-curved. The steel construction expresses regular diagonals in the facade of the building, is easier and quicker to assemble and is highly compatible with the concept of a sustainable building. In the design of the diagrid construction, an important factor is to choose the right diagonal angle. If the diagonal angle deviates from the optimum value, the required amount of steel is substantially increased. Since the optimum angle of placement of the columns for maximum bending stiffness is 90 degrees, and diagonals achieve maximum shear stiffness at an angle of 350, the optimum angle for diagrid construction elements is therefore taken between the values of these angles. The arrangement of diagonal elements with larger angles in the corners of the building increases its bending stiffness. High-rise buildings with a high ratio (height/width) behave like bent beams. Therefore, when a building's height rises, the optimum diagonal angle also increases. Buildings in this construction system are designed on a circle, ellipse, or other curved geometric form. The diagrid system is perfectly matched in the modification of the classical geometrical form. In this system, the following forms are known: Hyperboloidal, cylindrical, twisted, tilted and free [15].

### 3.3. Advanced Vibration Damping Systems

The development of the advanced damping system has been characterized on the basis of Japan, which has the most active seismic zone in the world and which paradoxically occupies third place in terms of the number of skyscrapers.

An essential aspect of designing tall buildings is their dynamic reaction to earthquakes and counteracting wind vortices. Moreover, high buildings are sensitive to wind-induced vibrations, and the impact of such vibrations becomes dominant for buildings higher than 200 m. Under the action of the wind, a building not only deflects statically from its vertical position in the direction of wind pressure, but can also fall into the vibrations, which are transverse to the wind direction. These vibrations become dangerous, and even resonant when taking place with the frequency of air vortexing from the sides of the building. A simple measure of the quality of a skyscraper, i.e., its resistance to oscillatory swaying and resonance with the wind, and also its dynamic stiffness under bending is the fundamental (minimum) natural frequency of the building. Possible oscillations form a spectrum of waves with different vibration frequencies—higher or shorter. However, more diverse waves occur

when the building bends. The longest wave with the smallest frequency, called the fundamental frequency, represents the shaking of the entire building. Such oscillations are most easily created at the lowest wind speed—hence their fundamental importance for the comfort of use and safety of the structure (risk of resonance with the wind). A better building has a higher fundamental frequency of vibrations, is stiffer, vibrates faster and introducing it into dangerous resonance vibrations requires a wind with a higher speed, which occurs less often and is less likely.

While the strength of building materials, such as steel, has doubled in the last few decades, its stiffness has not increased significantly. This has led to an elastic-based approach to design in which lateral deflections and accelerations are the dominant structural constraints for tall buildings. Vibrations can be partially damped by the structure itself. Increasing the stability of the structure causes an increase in the natural frequency. According to the numerical simulation of a construction's response to wind, if the natural frequency is greater, the maximum acceleration decreases in proportion to half of the natural frequency.

The light steel structure used in high-rise buildings has little natural damping or natural dissipation of energy and is sensitive to dangerous accelerations in conditions close to resonance. The dynamic reinforcement of load conditions can be reduced by redistributing stiffness in order to avoid resonance, or by the implementation of a damping system in the building (Figure 5). The need for motion control has led to the development of various methods and devices for dissipating energy. Damping devices can be passive, which do not require an additional energy supply, or active (AMD), which suppress the reaction with input energy, usually through the use of actuators [19,20]. Although there are many effective applications for active dampers, the increased complexity, maintenance and cost and lower reliability of passive dampers means that they are more often used. In addition to passive and active systems, there are mixed hybrid systems.

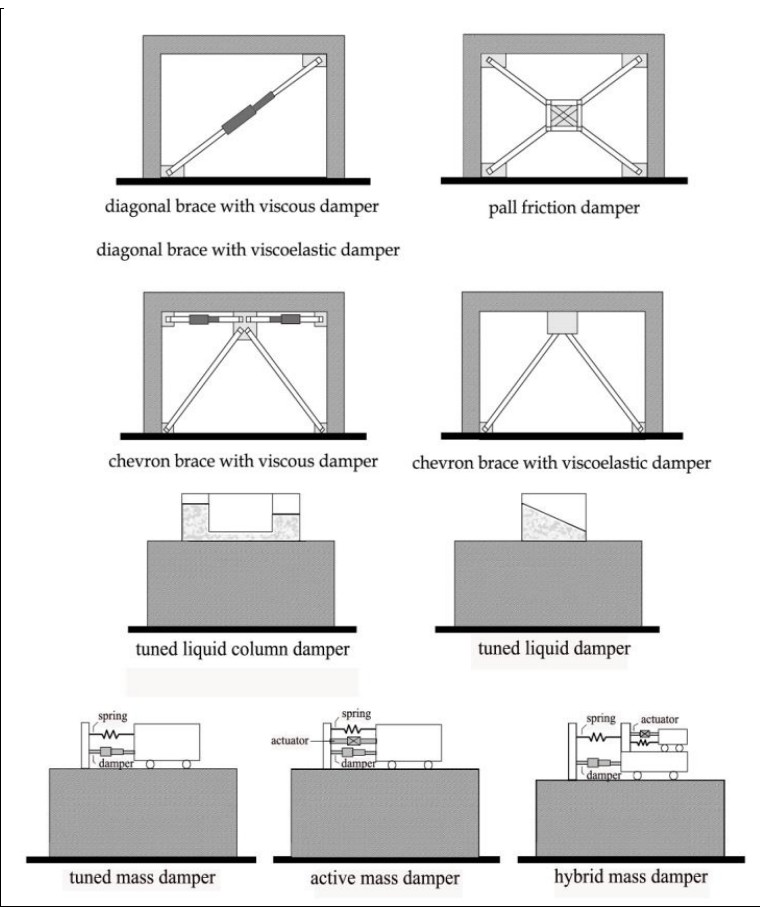

**Figure 5.** Vibration damping systems (figure by authors).

### 3.3.1. Passive Damping—Material Based Dissipation System

The material based dissipation dampers are an integral part of primary structural systems, and they are positioned in optimal locations (e.g., in bracing systems). There are different types of devices that belong to this category, and among all, the most important are: Hysteretic dampers and viscous dampers.

Hysteretic damping uses steel vibration absorbers SD (steel damper) and SJD (steel joint damper), as well as viscoelastic dampers (VED), lead dampers (LD) and friction dampers (FD), which are used to reinforce material interactions at the FD connections. Steel vibration absorbers dissipate energy through the cyclic inelastic deformation of materials. These damping systems are often designed in the form of a triangular plate, or are X-shaped. Due to this shape, plastic deformations appear in a much larger area, which leads to a more efficient dissipation of energy. This system was used in the Ohjiseishi Building (Tokyo, Japan), Art Hotels Sapporo (Sapporo, Japan) and Kobe Fashion Plaza (Kobe, Japan).

In friction dampers, energy dissipation occurs as a result of friction between two solids moving in relation to each other. There are two types of friction dampers used in steel framed buildings: Rigid frame friction dampers and braced frame friction dampers. For example, friction dampers were used in the Sonic City Office Tower (Ohmiya, Japan) and Asahi Beer Tower (Tokyo, Japan).

Viscous dampers (VD) and oleo-dynamic dampers (OD) use viscous materials in which the resistance force acting on the body moving in the material is proportional to the speed of the body [21]. In this case, high viscosity chemicals such as silicone oil are used. The thermal effect is also significant. VDs are particularly effective in the high frequency range and low vibration levels against moderate earthquakes and strong winds. This type of damper, consisting of steel plates, is installed as a part of a diagonal brace, where it can dissipate vibrational energy by the shearing action of the VE material.

Viscoelastic dampers were used in the TV-Shizuoka Media City buildings (Tokyo, Japan) and in the Torishima Riverside Hill Tower (Osaka, Japan) to counteract the vibrations caused by extremely large earthquakes.

### 3.3.2. Passive Damping–Additional Mass System

This passive system is based on the counteracting inertial force created by an additional mass allocated at the top of a building. There are two main categories of devices belonging to this group: Tuned mass dampers (TMD) and tuned liquid dampers (TLD). A TMD is an additional mass [22], usually in the order of two percent of the total weight of the building, which is attached to the structure by means of springs and dashpots. The inertia force of the mass damps the reaction of the building. However, TMDs are mostly effective only when they are excited by the resonant frequency for which they have been designed. Sometimes, spacing limitations do not permit a traditional TMD system, which requires the installation of alternative configurations such as pendulums, hydrostatic bearings or laminated rubber bearings. A TMD damper was used in Fukuoka Tower (Fukuoka, Japan), Higashimyama Sky Tower (Nagoya, Japan) and Huis Ten Bosch Domtoren (Nagasaki, Japan).

Another type of mass damping system is tuned liquid dampers (TLCD). This damping system uses the movement of liquids in special containers to absorb the energy of building vibrations. The vibration frequency of TLCDs can be controlled by the water depth and the size of the container. TLCDs are preferred because of their simplicity, low maintenance price and the possibility of including water for emergency fire protection. The TLCD system was used in the Rokko-Island P and G Building (Kobe, Japan), Crystal Tower (Osaka, Japan) and Sea Hawk Hotel and Resort (Fukuoka, Japan).

Different from passive systems that are tuned to work on some range of loading conditions, active system perform more efficiently over a wider range. The most prominent active devices are active mass dampers (AMD) and active variable stiffness devices (AVSD). Both devices rely on the same principles of mass and material based dissipation but their properties are adjusted from a computer control system. The AMD system was used in the Applause Tower (Osaka, Japan) and AVSD in ORC 200 Bay Tower (Osaka, Japan).

### 3.3.3. Hybrid Damping

In recent years, hybrid dampers have appeared, which are a combination of a mass damper with an additional active element, which aims to improve the efficiency of passive damping [23]. The forces from the active actuator increase the effectiveness of the mass damper and are very effective in the event of changes in the dynamic characteristics of a structure. The active portion of the system is only used under excitation of a high-rise building, otherwise, it behaves passively. The hybrid system was used in the Landmark Tower building (Yokohama, Japan), the Ando Nishikicho building (Tokyo, Japan) and in Osaka World Trade Center (Osaka, Japan).

### 3.4. Use of Advanced Materials

The development of high-rise buildings is inextricably linked to the search for efficient construction materials, Figure 6. Technological achievements in material engineering have gradually shaped the form, height and construction, as well as energy efficiency of buildings. Initially, steel was the leader in building constructions, as the technology of concrete was not sufficiently developed, and because the produced concrete had a much lower strength than steel. At present, there is a growing interest in concrete as the main structural material in this type of buildings [24]. In the construction of high-rise buildings are also developing mixed steel-concrete technologies, such as the Petronas Twin Tower (Kuala Lumpur, Malaysia), Burji Khalifa (Dubai, UAE), Princess Tower (Dubai, UAE), One57 (New York, NY, USA) and Kingdom Center (Riyadh, Saudi Arabia). Currently, among the 100 highest buildings in the world, nine are built as steel structures, 30 as reinforced concrete, 5 as steel and reinforced concrete and 56 as composite structures. Advances in physical science have led to a new generation of intelligent materials, especially those that improve the acoustic, light, electrical and thermal environment of buildings [25].

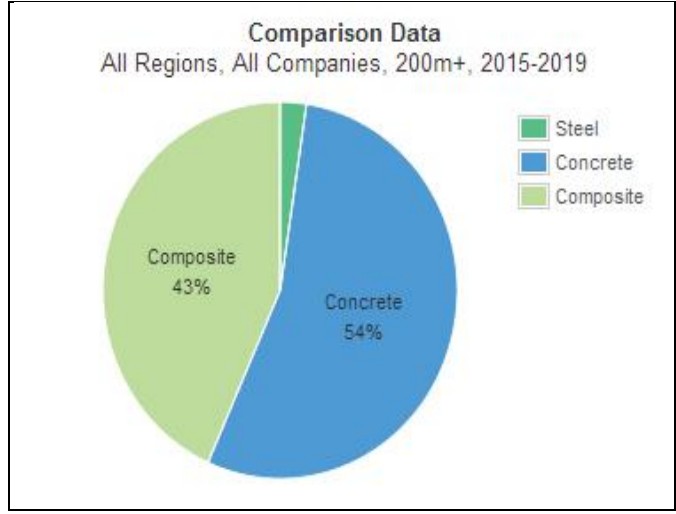

**Figure 6.** Comparison of material system applied in high-rise buildings (above 200 m) in the years 2015–2029 (on the basis of the global tall buildings database of CTBUH (Council on Tall Buildings and Urban Habitat)).

### 3.4.1. Concrete

Over the last years, there has also been significant progress in the field of modeling the physical and rheological properties of concrete. Added admixtures allow for a significant increase in strength, accelerate the curing of concrete and enable construction works at both very low and very high temperatures. Very high strength concrete (VHSC), self-consolidating (SCC) led the concrete to be the most appropriate structural material for super tall buildings, such as Burj Khalifa (828 m, Dubai, UAE) and Kingdom Tower (1000 m, Jeddah, Saudi Arabia). VHSC has a compressive strength of

240 MPa with steel fiber reinforcement incorporated in the mixture and achieves a flexular tensile strength of 40 MPa [26]. Moreover, the development of construction technology (moving formworks with high accuracy and speed of assembly and disassembly, vertical transport systems—pumps, etc.), high susceptibility to shaping, the faster growth in strength than prices, and high fire resistance are further advantages for the use of concrete. The development of concrete technology and methods of construction organization has not only allowed for the construction of higher and higher skyscrapers, but also for the diversification of their forms and shapes.

### 3.4.2. Steel

Despite the increase in the number of high-rise buildings made of ultrahigh-strength concrete, steel is still an irreplaceable material in seismic areas. Currently, Japan is one of the most advanced countries in terms of the development of steel structures. Despite the unfavorable geographical location associated with the occurrence of frequent earthquakes, Japan can boast of having such building structures as: Akashi Kaiyko bridge (one of the longest suspension bridges in the world, longest span 1991 m) and Tokyo Sky-tree (the highest free-standing tower, 634 m). Undoubtedly, the factor that influenced the creation of these structures was the improvement of the efficiency of steel materials, which favored their development. TMCP (Thermo-Mechanical Control Process) technology was used to obtain high-strength steel. This technology is a combination of "controlled rolling", which favors the refining of the microstructure by introducing dislocation in a high temperature range, and also "accelerated cooling", which realizes the quenching effect while suppressing grain growth. With such limited carbon technology, high-performance steel materials with excellent weldability and efficiency can be produced. The use of high-strength steel in high-rise buildings was a consequence of their earlier use in bridge constructions. At present, steel with a tensile strength of over 1200 N/mm$^2$ is available [27]. This high strength was achieved by the development of a dual-phase steel (DP steel), which has a structure composed of hard and soft material and TRIP steel in which the plasticity effect of unstable austenite is caused by martensitic transformation. In addition to the tendency to increase the strength of steel, there was also a demand for steels with a low yield stress (yield strength 100 N/mm$^2$ class and yield strength 225 N/mm$^2$ class), which were first used for the construction of vibration dampers. Since the plastic deformability of high strength steel is lower than for conventional steels, the performance of an entire building is achieved with a combined use of dampers.

### 3.4.3. Smart Material and Nanotechnology

Smart materials can be divided into the following groups: Piezoelectric, electroactive, photostrictive, thermostrictive, magnetostrictive, chemostrictive materials and fiber optic sensors [28]. These smart materials can constitute the components of a smart structure, which is an electronically enhanced physical framework. For example piezoelectric materials convert mechanical energy into electrical energy after strained. Piezoelectric dampers have been developed as an example of controllable materials. There are other forms of smart materials, such as shape memory alloys, which can be used as temperature sensors for ventilation systems or as actuators for sensing and monitoring devices. With nanotechnology can be improved properties of glass by self-cleaning, antimicrobial and reducing pollution properties. Titanium dioxide nanoparticles with a smooth surface create an anti-adhesive coating.

### 3.4.4. Glass

Technologically advanced high-strength glazing is equally important as steel and concrete for the building of high-rise buildings. In this case, the main challenges are related to wind load, temperature and altitude differences, and also the condensation of water vapor. Other important factors are light and heat. In the case of high-rise buildings, there is always the possibility of condensed steam appearing on the outside glass, which results from the temperature difference between its internal and external part. The use of low-emission glass as an internal pane prevents the passage of heat from inside the building

to the outside. Low-E glass helps to reflect long-wave radiation and minimizes its transmission. Heat treatment of the glass through hardening or heat strengthening causes the glass to be many times stronger and able to withstand extreme wind load and temperature difference. In high-rise buildings, a wide variety of glass types are used depending on the climate zone, Table 1. To fully characterize glass system, it is necessary to specify the following characteristics: U-value, solar heat gain coefficient (SHGC) and glass visible transmittance [29,30].

**Table 1.** Indicative characteristics of different glass types [30].

| Glass Type | Glass Thickness (cm) | Visible Transmittance (% daylight) | U-Factor (Winter) | Solar Heat Gain Coefficient |
|---|---|---|---|---|
| Single Pane | 0.63 | 89 | 1.09 | 0.81 |
| Single White Laminated | 0.63 | 73 | 1.06 | 0.46 |
| Double Pane Insulated | 0.63 | 79 | 0.48 | 0.70 |
| Double Bronze Reflective | 0.63 | 21 | 0.48 | 0.35 |
| Triple Pane Insulated | 0.32 | 74 | 0.36 | 0.67 |
| Pyrolitic Low-e Double | 0.32 | 75 | 0.33 | 0.71 |
| Soft-coat Low-e Double | 0.63 | 73 | 0.26 | 0.57 |
| High Efficiency Low-e | 0.63 | 70 | 0.29 | 0.37 |
| Suspended Coated Film | 0.32 | 55 | 0.25 | 0.35 |
| Suspended Coated Film Argon gas fill | 0.32 | 53 | 0.19 | 0.27 |
| Double Suspended Coated Film | 0.32 | 55 | 0.10 | 0.34 |

U-value indicates the rate of heat flow due to conduction, convection and radiation through a glass as a result of the temperature difference between the inside and outside. The higher the U-factor the more heat is transferred through the window in winter.

SHGC indicates how much of the sun's energy striking the glass is transmitted through the glass as heat. As the SHGC increases, the solar gain potential through the window increases.

Visible transmittance indicates the percentage of the visible portion of solar spectrum that is transmitted through a glass.

*3.5. Innovative Energy Systems in High-Rise Buildings*

The achievement of high energy efficiency in modern high-rise buildings requires many environmental conditions to be taken into account at the stages of design and construction. Satisfying these requirements allows the maximum use of available ambient energy, the reduction of heat loss from the building, and also a smaller demand for heat and electricity. One of the most finance-intensive requirements is the ventilation and heating of buildings, accounting for about 30% of the energy demand in high-rise buildings. The use of natural ventilation is an increasingly popular solution that reduces these costs. The inner atrium allows light to be supplied to the interior of the building, Figure 7. The full height of the windows causes the amount of light reaching inside to be sufficient for work, and there is therefore no need to use artificial lighting for most of the day. The ventilation of the rooms is also ensured by specially designed windows, constructed of a three-layer facade system with an air gap, allowing air to circulate.

The use of free energy from renewable sources, such as sun, wind, biomass and low-temperature geothermal energy, is also becoming more and more popular. This is especially the domain of passive buildings, and also sometimes of energy-saving buildings. Among the activities preceding the implementation of a project, the selection of the right location is of particular importance and

results in the efficient use of available renewable energy sources. The next elements are: Adaptation of the architectural design to local microclimatic conditions, proper location of the building, accurate orientation towards the sun and correct shaping of the surroundings of the nearest building. The location of buildings should provide good insolation conditions and the maximum number of hours of sunshine per year.

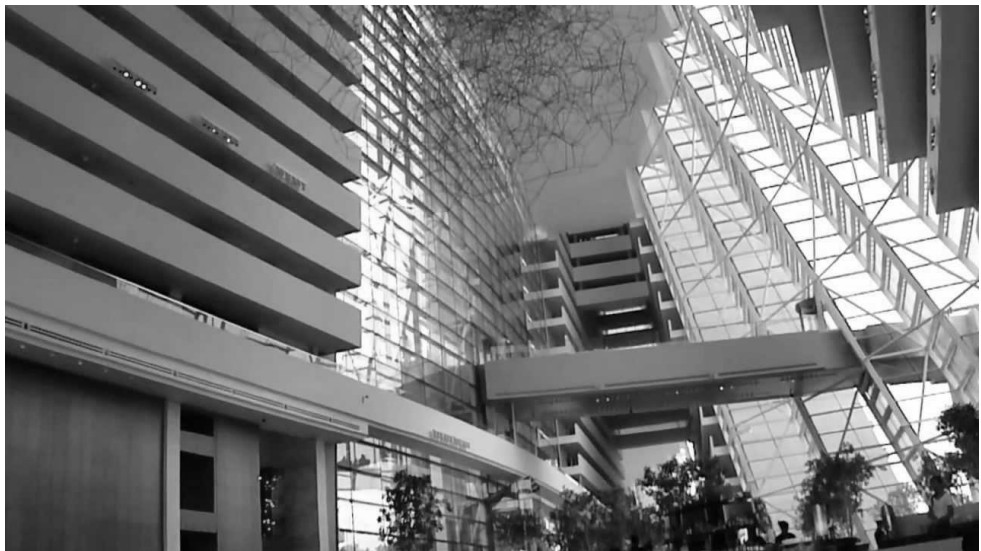

**Figure 7.** Atrium in the Marina Bay Sands Hotel (Singapore, photograph by authors).

Such a situation is beneficial for bioclimatic reasons, as well as for the possibility of using solar energy in active and passive photothermal and photoelectric conversion systems. Direct conversion includes:

- Photothermal methods, implemented in low-temperature active solar systems (solar collectors, Figure 8) and in passive systems (solar architecture of buildings) [31],
- Photoelectric methods, implemented in photovoltaic systems (cells) [32], Figure 9.

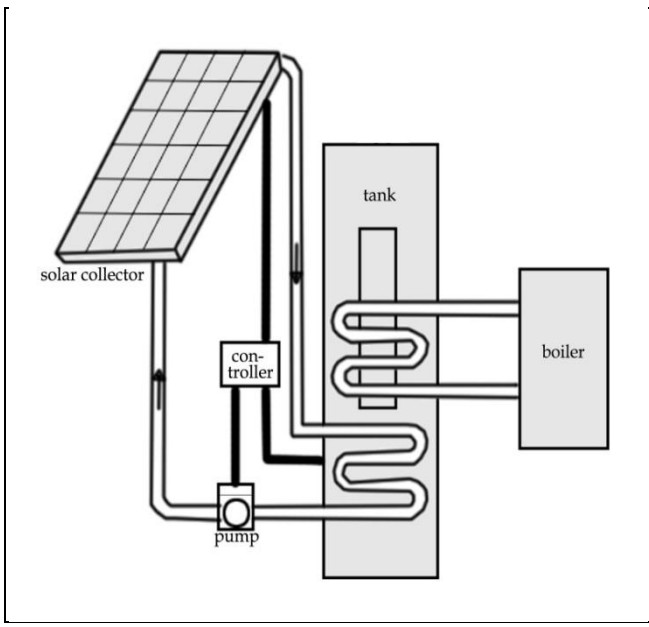

**Figure 8.** Photothermal solar technology (figure by authors).

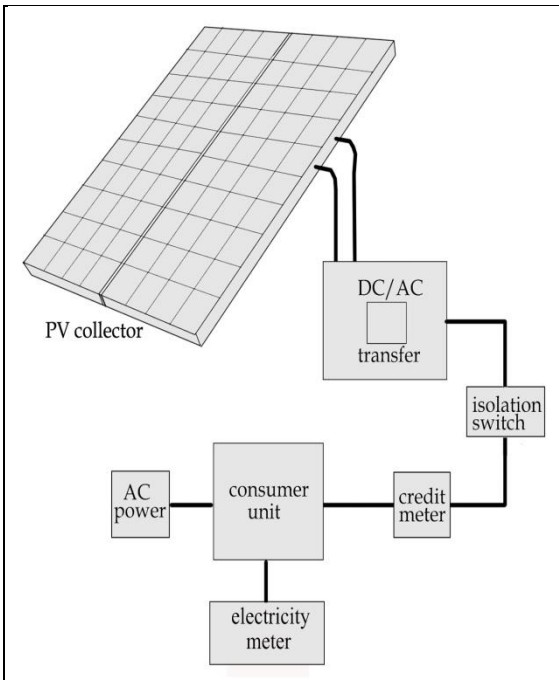

**Figure 9.** Gallium arsenide photovoltaic cells technology (figure by authors).

In the active solar system, energy acquisition and also its separation and storage takes place through the use of such elements as: Solar collectors, storage tanks, safety devices and elements of control and measurement automation. In addition to the proper selection of system components, it is important to properly arrange the collectors by setting the right angle of deviation from the southern direction (declination), and also the angle of inclination with the ground plane (inclination). Passive solar heating systems for generating heat in a building use solar radiation energy directly or indirectly. To achieve a more sustainable design gallium arsenide photovoltaic cells combined with a rain screen in the southeast facade are often used [33].

*3.6. New Technologies of Facades*

One of the most significant changes in technical solutions and the aesthetics of high-rise buildings was caused by the role of the contemporary glass facade of the building. Architects Norman Foster and Thomas Herzog [34] in the European Charter for Solar Energy in Architecture and Urban Planning stated that the building's exterior walls in terms of light, heat, air and transparency must be susceptible to change and ultimately be controllable to respond to changing local climate conditions. It is noticeable in the last few years of development of new advanced facade solutions integrated with plants to combine architectural features and trends to reduce carbon emissions. The concept of vertical gardens in the form of green facades of buildings is now a trend of sustainable design, in which the ecological facade material offers an unlimited number of patterns and colors that change both during the day and in different seasons (Oasia Hotel downtown (Singapore). The presented structure, in the form of a wall covered with plants, aroused the interest of architects and finally resulted in cooperation. Numerous ecological green designs were implemented by Jean Nouvel (One Central Park (Sydney, Australia)), Herzog and Pierre de Meuron (Beirut terraces (Beirut, Libanon)) and Stefano Boeri (Bosco Verticale (Milan, Italy), Nanjing Tower (Nanjing, China)).

Currently ventilated double skin facades represent a most valid technology (Figure 10) [35,36]. The principle of ventilated double skin facade is to position the shading devices between two layers of glazing, capturing the energy trapped in the cavity (Figure 11). Among the technologically advanced facades, it can be distinguished by two types: Active wall facade (Manulife Financial, Boston; Jiu Shi Headquarters, Shanghai) and interactive wall facade (Al Bahr Towers, Abu Dhabi).

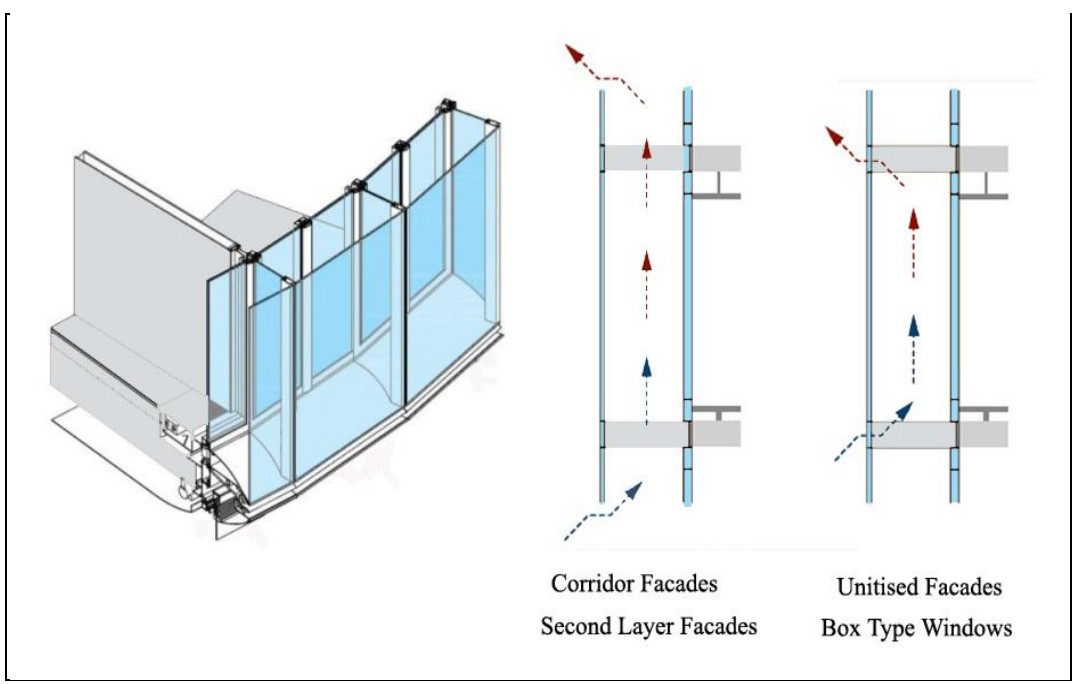

**Figure 10.** Natural ventilation of double skin facades (figure by authors).

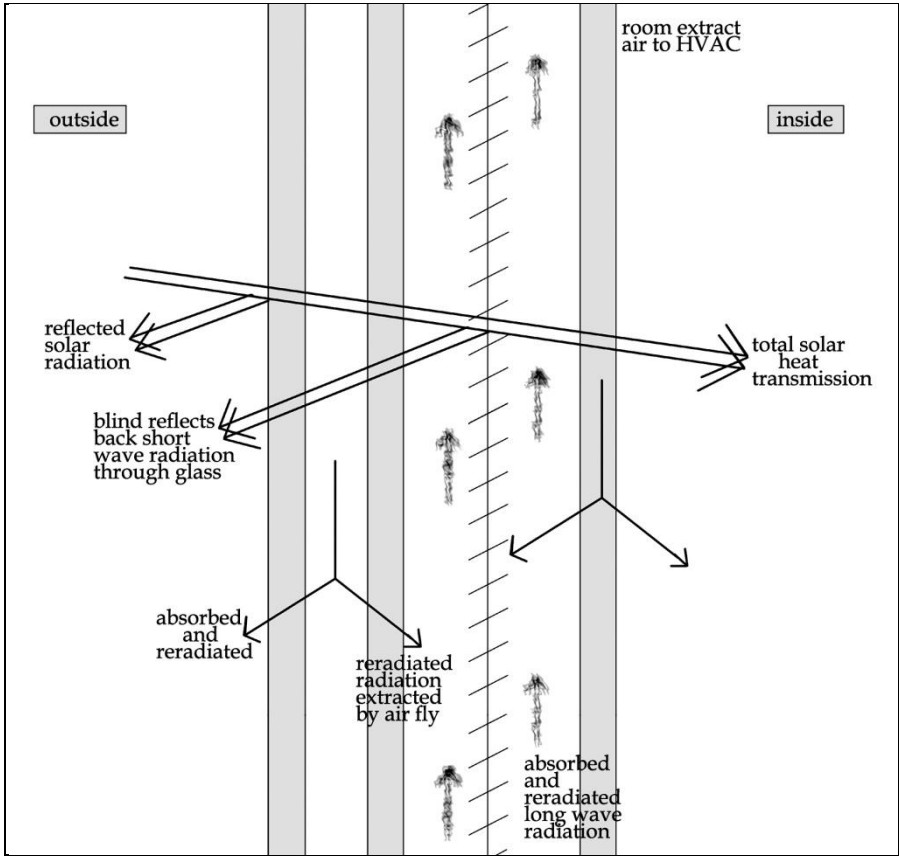

**Figure 11.** The active facade system (figure by authors).

The system performance depends on numerous design parameters (light transmission, solar factor, thermal transmittance and acoustic insulation), which values for different facade typology are shown in Table 2. The integration between double facade and environmental systems generally results in a reduction of installed power for heating and cooling due to a reduced U-value, a lower solar factor

for hot climate (SHGC) and a potential heat recovery. The solar factor of the facade is a key variable to control both the reduction of overall energy and separate the cooling demand from the orientation of the building.

**Table 2.** Indicative characteristics of different facade typology [37].

| Façade Typology | Light Transmission $\tau_{vis}$ (-) | Solar Factor SHGC$^2$ (-) | Thermal Transmittance U–Value (W/m$^2$K) | Acoustic Insulation $R_w$ (dB) |
|---|---|---|---|---|
| Externally shaded | 0.50–0.70 | 0.05–0.25 | 2.0 | 32–36 |
| Naturally ventilated | 0.60–0.70 | 0.10–0.20 | 1.4 | 38–44 |
| Active Wall | 0.60–0.70 | 0.15–0.25 | 1.0 | 38–44 |
| Interactive Wall | 0.60–0.70 | 0.10–0.20 | 1.3 | 38–44 |

High-rise buildings with traditional glazed curtain walls that allow sunlight to penetrate indoors cause unwanted heat gains and losses, thereby increasing cooling or heating. In a naturally ventilated facade, the cavity between the two skins is ventilated with outdoor air. An active wall facade is composed of an external insulating glazing unit and an internal single layer of glass. The cavity between the two skins is ventilated with return room air, which is extracted from the room at the base of the glazing and returned to the air-handling unit at the top.

Interactive wall facade has a digital, mechanical adaptive solution system, which can react. The concept is that the interior side of the facade interacts with its inhabitants, perceiving their body movement and adjusts its form accordingly. The exterior side interacts with the movement of the sun, working as a shade, reacting to environmental changes.

In some type of interactive wall facade, the cavity between two skins is ventilated with outdoor air at the base of glazing and returned to the outside at the top by means of temperature-regulated radial fans located in the upper part of the facade.

To manage sunlight appropriately and provide occupant comfort, innovative window systems and glazing are developed in order to regulate the sunlight. Different types of glass and film coatings, such as low-E, are used to enhance the performance of the façade [38].

At present, systems where the air exchange is limited to the height of one story are considered optimal, and this serves to prevent the possibility of the release of used air into rooms located on higher floors. For this purpose, glass ribs are used to separate the individual sections of the facade vertically. An integral part of the contemporary double skin facade is a sunblind placed in the space between the glazing layers. Its task is to reduce the penetration of direct sunlight into the rooms so as to reduce the amount of heat accumulating in them. This new generation of high-performance envelopes have contributed to the emergence of sophisticated assemblies that combine a real-time environmental response, advanced materials, dynamic automation with embedded microprocessors, wireless sensors and actuators and design-for-manufacture techniques.

## 4. Examples of Selected High-Tech High-Rise Buildings

### 4.1. Burj Khalifa (Dubai, UAE)—The Highest Mega-Structure System

Burj Khalifa is a mixed use skyscraper with a steel and reinforced concrete structure designed by the architectural studio Skidmore, Owings and Merrill. The building is 829 m high and contains 163 floors at the above-ground level and one underground floor, Figure 12. The initial concept for geometry of the Burj Khalifa originated from the hymenocallis flower, characteristic element in Islamic architecture. The form was designed based on how to counteract the effect of the wind on the structure. Burj Khalifa is designed on Y-shaped plan (Figure 13), which is ideal for residential usage, allowing maximum outward views and inward natural light.

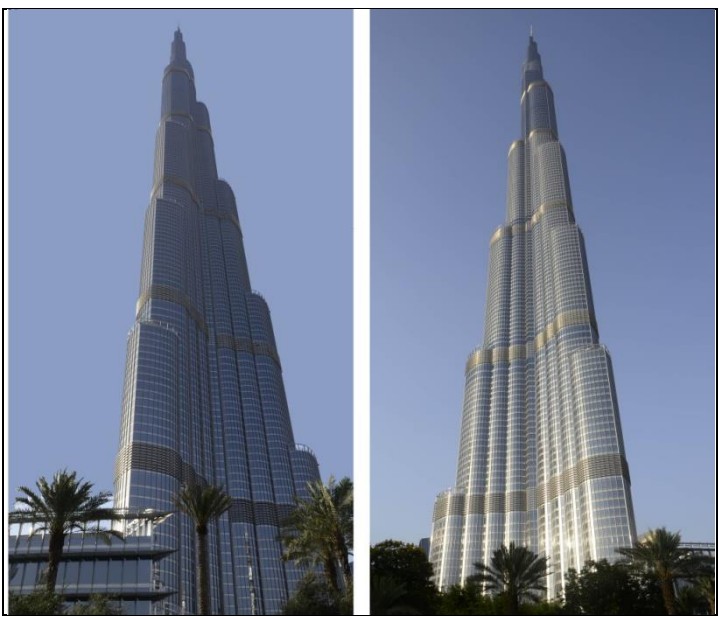

**Figure 12.** Burj Khalifa (photograph by authors).

To achieve a very strong and stiff building the latest technology in materials, analysis and construction methods were used. The skyscraper has a raft foundation that is 3.7 m thick, and cooperates with 192 piles that are 47 m in length [39]. A cathodic protection system creates a barrier to protect structure from polluted ground water affecting the concrete [40].

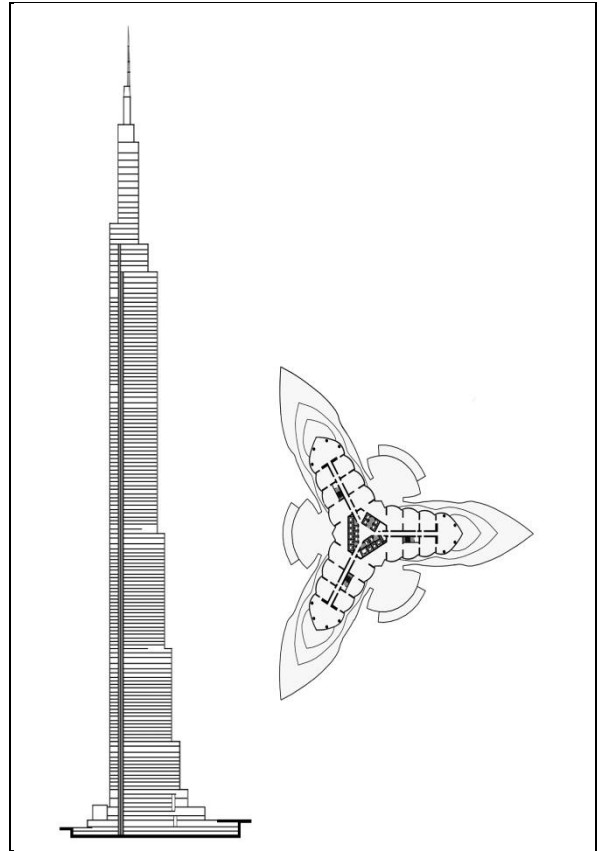

**Figure 13.** Burj Khalifa: Floor plan and section (developed by authors on the basis of [41]).

The main structural system for this mega-structure is the buttressed core, which allows a very big increase in height. The structural system consists of three-winged structure anchored to a hexagonal central core. The central core provides the torsional resistance for the building, and wings provide the shear resistance and increased moment of inertia. With the addition of floor RC (Reinforced Concrete) slabs and perimeter columns, the entire structural system acts like a single unit creating the tower.

The skyscraper is set on the podium, which provides a base anchoring the structure to the ground, allowing a grade access from three different sides to three different levels. The steel spire crowns the building has a diagonally braced lateral system.

The exterior cladding is comprised of reflective glazing with aluminum and textured stainless steel spandrel panels (Figure 14) and stainless steel vertical tubular fins. The cladding system is designed to withstand Dubai's extreme temperature, and to further ensure its integrity.

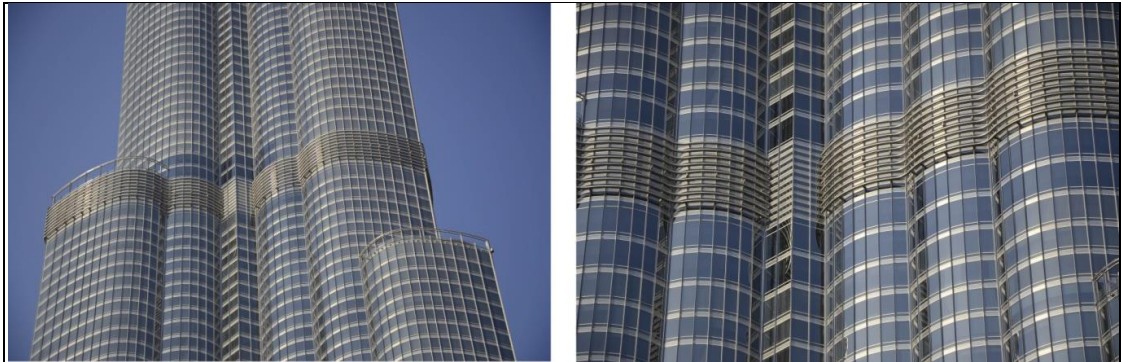

**Figure 14.** Burj Khalifa: Facade-reflective glazing with aluminum and textured stainless steel spandrel panels (photograph by authors).

Fire safety and speed of evacuation were prime factors in the design of Burj Khalifa. The skyscraper was the first mega-high rise in which certain elevators are programmed to permit controlled evacuation for fire or security events.

Burj Khalifa is a pro-ecological building with many innovative technological solutions. The solar heating system located on the roof of the offices is installed, which serve as solar collectors. Among other key sustainable energy and water use, the condensate from all the air-conditioning equipment is reclaimed to cool the potable water. The condensate is then collected in an on-site irrigation tank and used for the skyscraper's landscaping.

### 4.2. Sky Tree Tower (Tokyo, Japan)—Advanced Damping System

Tokyo Sky tree is a radio-television and observation tower with a steel and reinforced concrete structure as shown in Figure 15. The tower is the highest in the world with a height of 634 m and was designed by the architectural studio Nikhen Sekkei. The tower has two observation areas, the Tembo Deck and the Tembo Gallery. Tokyo Sky tree employs advanced technologies and never before adopted design approaches. Configuration of towers seems simple, but in actuality it contains extremely complex curves. The building at the base was designed on a triangular plan, which progressively changes from a triangular shape to a circular form higher up, Figure 16. This unique configuration does not occur anywhere else in the world.

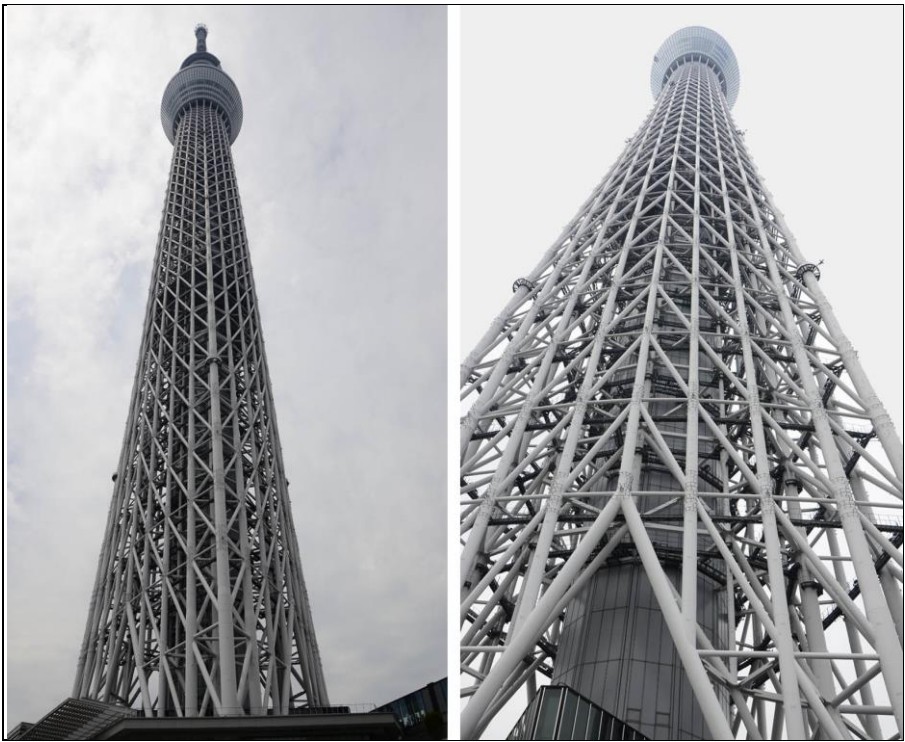

**Figure 15.** Tokyo Sky tree (photograph by authors).

Tokyo Sky tree is located on the banks of the river, where the surface layer is soft silt. The foundation of the tower consists of steel piles filled with concrete, and also reinforced concrete walls with a thickness of 1.2 m that are located at a depth of 35 m on the load-bearing layer under the surface of soft silt. A set of cylindrical steel and thin-walled piles reaches up to a depth of 50 m [42]. This system of rigid foundation construction and vulnerable ground uses a relative displacement that is used to damp vibrations. For the tower to be able to withstand the uplift and compressive forces of earthquakes and strong winds, piles have nodule-protrusions that work to hold the piles firmly in the ground, greatly increasing their strength in supporting the tower. The foundation must not only ensure horizontal stiffness, but also vertical stiffness, as well as counteract the overturning moment.

The tower's structure consists of two separate parts, one of which is a steel truss, the other an internal reinforced concrete core. Both parts can move independently. To minimize seismic energy, a central core or so-called shin-bashira [43], utilized for centuries in traditional Japanese architecture in pagodas, was used. The core has a diameter of 8 m, a thickness of 6 m, a height of 375 m and operates on a stationary pendulum that balances seismic waves by reducing vibrations. Additionally, the elements supporting the reduction of vibrations are viscous oil dampers attached to the upper part of the core [44]. An independent steel truss structure employing wide-bore high strength steel pipes was used, which can be rarely found in building construction, Figure 17a. The largest section applied at the foot of the tower is 2.3 m in diameter, made from 10 cm thick steel plates. The truss is not only light and strong, which is necessary from the design point of view in seismic areas but is also effective in a wind-resistant construction, reducing the frontal area and not causing an unstable aerodynamic reaction due to the absence of an external wall. There are two types of steel structure in the tower. One structure is a truss, and the other one is a mega truss with a lattice core and girder, known as the Kanae truss [18]. These trusses are composed of four main members and are located in each corner of the equilateral triangle of the tower base. The three-level Tembo Deck features an observation floor that is situated at height 350 m. The Tembo Gallery has two floors, connected to each other by a circumferential ramp, Figure 17b. The higher of these floors is 451 m above the ground.

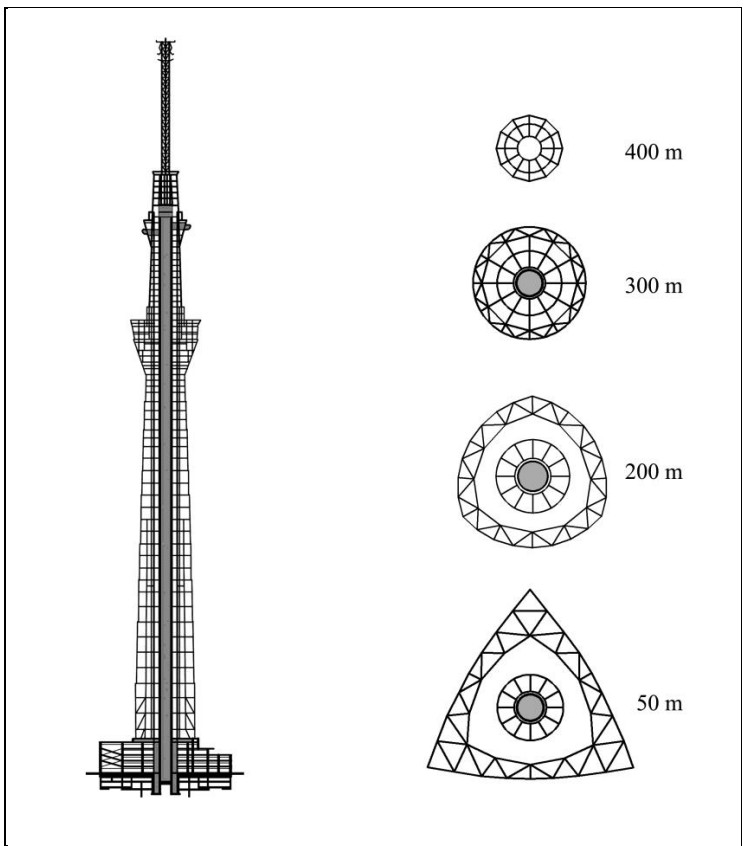

**Figure 16.** Tokyo Sky Tree: Structural outline (developed by authors on the basis of [44]).

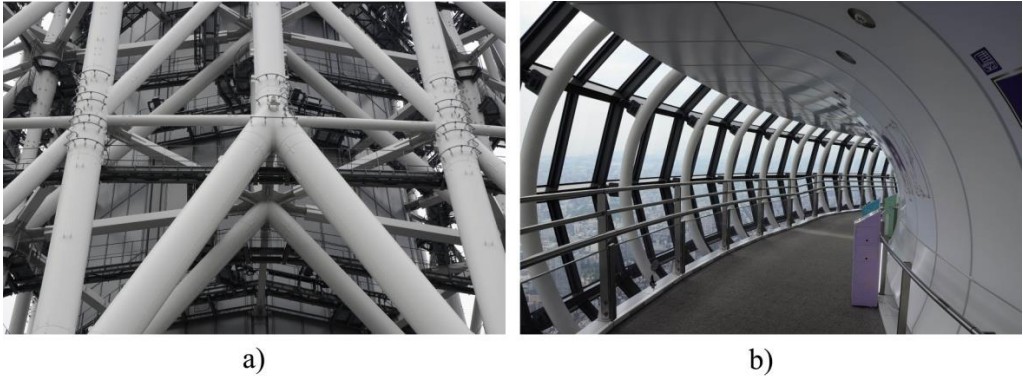

a)                                   b)

**Figure 17.** Tokyo Sky Tree: (**a**) An independent steel truss structure employing wide-bore high strength steel pipes; and (**b**) the spiral ramp of the Tembo Gallery in the second observation deck (photograph by authors).

To minimize the impact of wind in the upper part of the tower, a system of tuned mass dampers was installed. In practice, this is characterized by two massive ballast weights, weighing 25 and 40 tons, which were supplied by Mitsubishi Heavy Industries and hung close to the top with large springs and vibration absorbers. As in the case of structures with a reinforced concrete core and an external truss, these two counterweights work on shifting any lateral movement.

*4.3. Capital Gate (Abu Dhabi, UAE)—Complex Geometrical Form and Advanced Structural System*

Capital Gate Tower is a mixed use high-rise building (hotel and office) with steel and reinforced concrete structure an original very complex geometric form, designed by architectural studio RMJM

(Robert Matthew, Johnson Marshal). The building is 164.7 m high and contains 36 floors at the above-ground level and one underground floor, Figure 18.

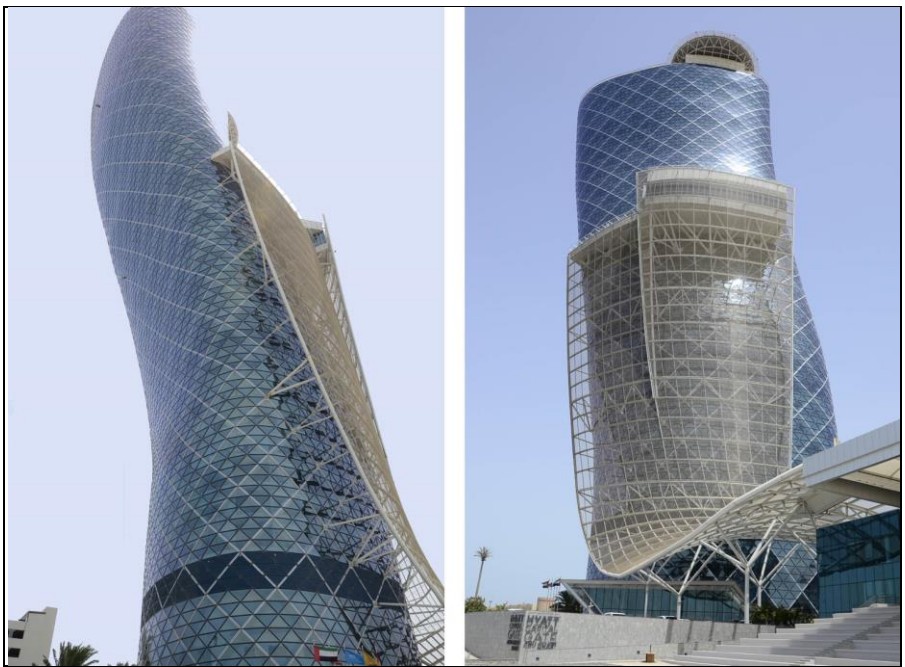

**Figure 18.** Capital Gate Tower (photograph by authors).

The geometrical form (Figure 19) is meant to represent a swirling spiral of sand, while the curved canopy that runs over the adjoining grandstand creates a wave-like effect reflecting the building's proximity to water and the city's sea-faring heritage.

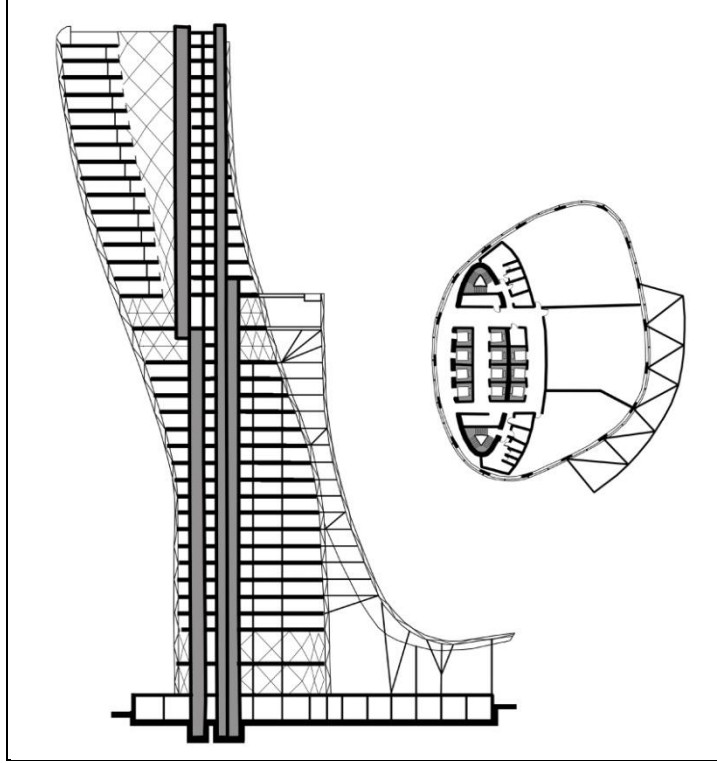

**Figure 19.** Capital Gate Tower: Floor plan and section (developed by authors on the basis of [45]).

Capital Gate has a raft foundation that is 2 m thick, and cooperates with 490 piles and drilled 30 m underground to withstand wind, seismic and gravitational forces caused because of its inclination [45].

The tower is a two-layer design. Capital Gate's base structure is a vertical concrete core surrounded by an advanced steel diagrid that determines the external form of the tower, Figure 20.

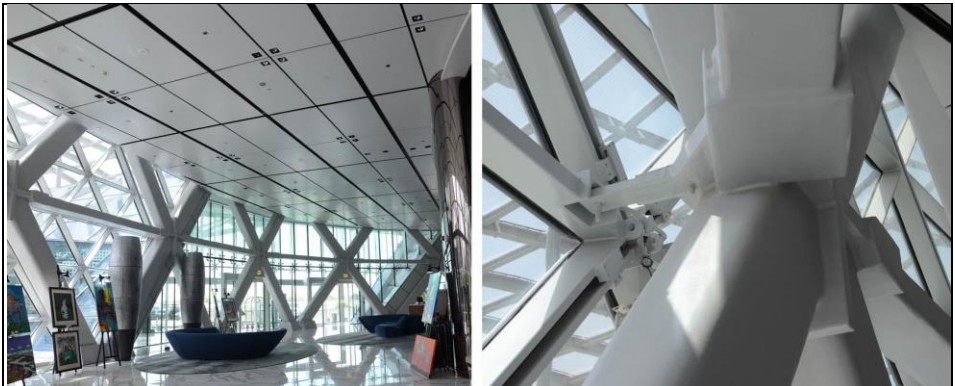

**Figure 20.** Capital Gate Tower: Diagrid structure—view from the interior (photograph by authors).

On the ground floor, a massive concrete ring beam transfers the thrust of the diagrid into the foundations. The central pre-cambered core leans in the opposite direction to the inclination of the building and straightens with the height. The building inclines by 18 degrees and holds the record for the furthest leaning tower in the world. A triangular structure with a diagonal support beam is applied for the formation of the diagrid.

Steel beams span between the two, and support metal deck and concrete composite floor slabs. An 80 m long cone atrium is formed above the base with an internal steel diagrid attached to the core. The post-tensioned core was designed with vertical cables on one side that are tensioned to counteract the lean on the other side. Steel girders span directly between the external and internal diagrids. The Capital Gate's wind bracing is designed as a separate system.

Capital Gate's most visible sustainable feature is geometrical twist around the building towards the south, to shield the building from direct sunlight. The double skin facade is made of glass panels and steel elements, which are matched to the curvature and form a diamond shape. The special glass minimizes the intense summer glare through the use of ant-glare elements in two silver coatings.

### 4.4. One World Trade Center (New York, NY, USA)—The Most Safe Building Structure

One World Trade Center is an office tower of reinforced concrete and steel structure designed by David Childs from the architectural studio Skidmore, Owings and Merrill. The building is 541 m high and contains 94 floors at the above-ground level and five under-ground floors, Figure 21. The first floor is occupied by a spacious atrium with a height of 15 m. The skyscraper is topped with a 124 m high spire [46]. There are antennas serving as a broadcasting tool for radio and television transmission in the ring of the spire. The One World Trade Center has a rectangular body with cut corners based on a rectangular base. As the height of the building increases from the base level, its edges form a geometrical form, which consisted of eight elongated isosceles triangles (four up and four down, alternately) [47]. In the middle part, the plan has the form of an ideal octagon. Then it is topped with a glass attic, which in the plan has a square shape with a size of 45 m and is turned 45 degrees to the base. The form of the building refers to the shape of the crystal and similarly to it, breaks the sun's rays.

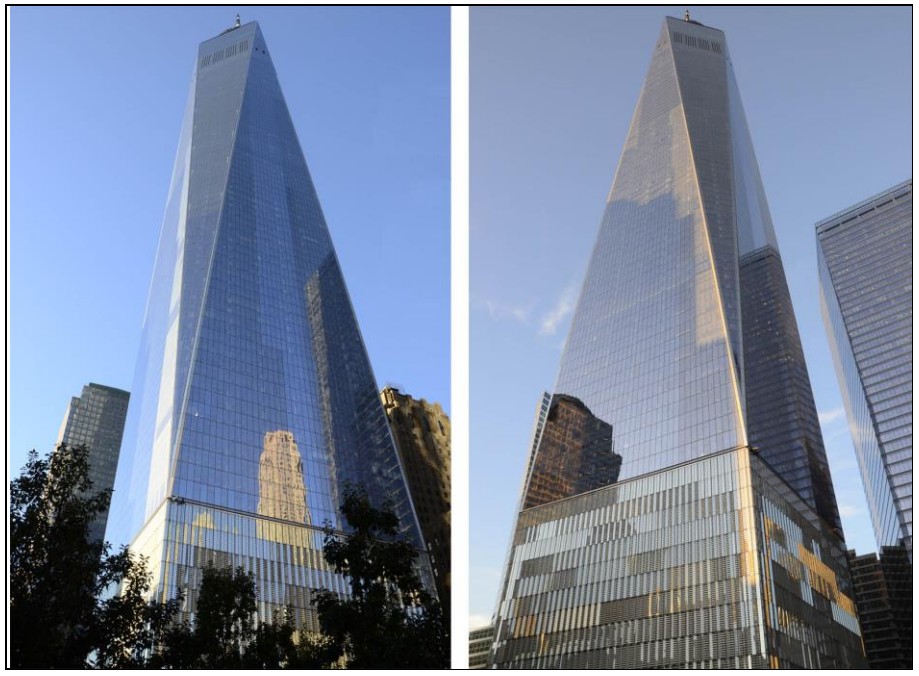

**Figure 21.** One World Trade Center (photograph by authors).

The base of the skyscraper is designed on a square plan (61 m), whereas the structure is a square with cut corners. In the central part there is a reinforced concrete core on a square plan (33.5 m) in which staircases and elevators are placed, Figure 22. The reinforced concrete base has a height of 19 floors, Figure 23. From 20 floors, the plan of individual floors and cross-sections of the core change with the shape of the body.

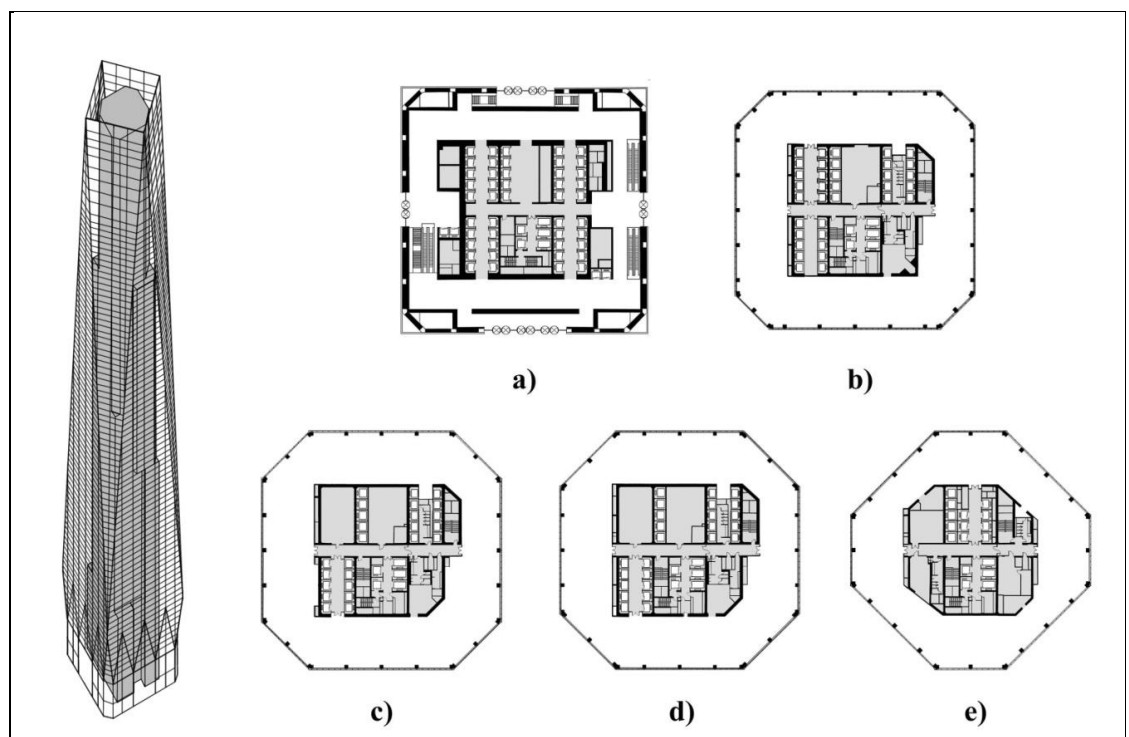

**Figure 22.** One World Trade Center: Main structure and floor plans: (**a**) Ground plan, (**b**) 45–49 floors, (**c**) 56–59 floors, (**d**) 60–63 floors and (**e**) 80–89 floors (developed by authors on the basis of [46]).

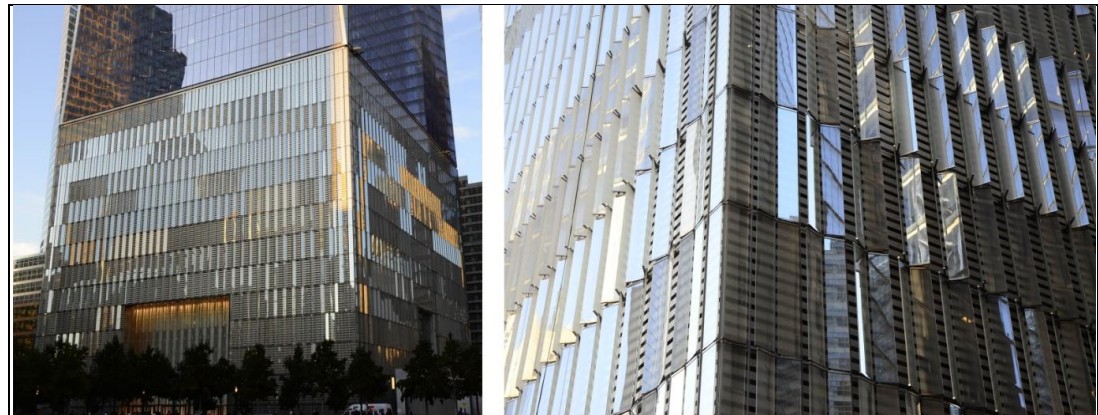

**Figure 23.** One World Trade Center: Reinforced concrete base with aluminum claddings and glass panels mounted in special profiles (photograph by authors).

The One World Trade Center tower was founded in granite rock with the use of long strip footings and spot footings with a capacity of six tons per square meter. Due to spatial constraints caused by the proximity of existing railway lines, a deeper foundation was necessary to obtain a higher load capacity. The anchorage of the foundation in the rock reached a depth of 24 m to counter the effect of the overturning moment as a result of extreme wind action.

The geometric shape of the One World Trade Center, with the body narrowing together with the height, in combination with cut corners effectively reduces the impact of the wind [47]. The structure of the tower consists of a hybrid system combining a massive reinforced concrete core with a peripheral steel frame [48]. An important element of the construction is a nineteen-story reinforced concrete base, whose massive reinforced concrete walls serve as a hidden safety barrier.

The reinforced concrete wall core in the middle of the building is the main supporting element that carries gravitational loads and counteracts horizontal loads from the wind and seismic effects. Due to its very high stiffness, there was no need for a special vibration damper. The core has a square plan with a length of 33.5 m, which is enough to constitute an independent building. The slab floor system without a column extends between the core and the peripheral steel frame.

Ultra-high-strength concrete used for thick concrete walls of the core, referred to as mass concrete, required a concrete mix to fulfill the most stringent requirements. All mixtures, depending on the needs, contained additional cement materials, fly ash, granulated lump slag cement and silica fume [49]. Due to the high height and slenderness of the building structure, the proportions of the concrete mix for the core were designed taking into account creep, shrinkage and modulus of elasticity.

To increase the safety of the building, designers used much less steel in the construction, and more composite materials. The foundation was made of concrete, but with an admixture of a substance that increases resistance to shocks (green concrete [50]). The 56 m high base is a windowless concrete wall designed to absorb a shock wave from a possible explosion. In building were applied triple laminated Viracon glass, whose high-performance coating reduces the amount of penetrating heat, UV radiation and infrared, while maintaining maximum visible light transmission.

Each lift in 1WTC (One World Trade Center) is protected in the central structure of the core, which is basically a vertical concrete bunker. Partition walls are reinforced with concrete and non-combustible materials. The concrete, which was used, is practically fireproof and the stairwell is subjected to pressure to prevent smoke from entering escape stairs.

The ventilation system was secured with special filters in the event of a terrorist attack using chemical or biological substances. There is also a special staircase for rapid response, used by rescue teams when public stairs are not available. Separate water and fireproof elevators are provided for firemen and security personnel [46]. All escape routes, e.g., stairs, have independent systems: Ventilation, radio and lighting. Air quality controls three thousand sensors. If one detects an excessive amount of carbon dioxide, it immediately transmits a signal to the computer, which automatically

pushes more oxygen into the room. In the case of a fire, the building has double-capacity water tanks in relation to the standard requirements for New York buildings. In addition the sprinklers and emergency call buttons are protected by concrete shields.

The One World Trade Center is one of the greenest office buildings in the world, belonging to the fifth generation in terms of energy consumption [51]. Over 30% of the used materials come from regional sources and 25% from post-industrial recycled materials. The building is partially secured by the supply of energy through 12 hydrogen-powered batteries. They generate 4.8 MW of power for themselves and other buildings in the complex. One World Trade Center incorporates not only new architectural and safety standards but also new environmental standards setting a new level of social responsibility in urban design.

### 4.5. Bahrain World Trade Center (Manama, Bahrain)—Advanced Sustainable Building with Large-Scale Integration of Wind Turbines

Bahrain World Trade Center is a complex of two twin office towers (Figure 24) with a reinforced concrete and steel structure designed by Architectural Studio Atkins. The buildings have an intelligent and environmentally responsive design. The two towers are 240 m high and contain 45 floors at the above-ground level and one underground floor, as shown in Figure 25. The towers have the shape of a sail and support three wind turbines (Figure 26) with a diameter of 29 m, which are supported on three different levels by bridges stretching between them [52]. The Bahrain World Trade Center is the world's first large-scale integration of wind turbines into a building. The towers are integrated on top of a three-story podium and basement. Each tower has a separate continuous piled raft foundation. The raft slabs have a different thickness according to loading and also incorporate lift pits. The raft thickness is 3 m beneath the main cores and the piles are 1.2 m in diameter. Away from the main core, the raft thickness reduces progressively to 2 m, and the piles to 1.05 m. The primary structure comprises two reinforced concrete cores. The main core houses lifts, escape stairs, plant rooms and toilets, and the secondary core houses escape stairs for the MEC (Mechanical) rooms. The floor plates typically have a story height of 3.6 m and are framed with reinforced vertical concrete columns on an 8 m grid and raking columns, which follow the sloping face of the building as it tapers in elevation.

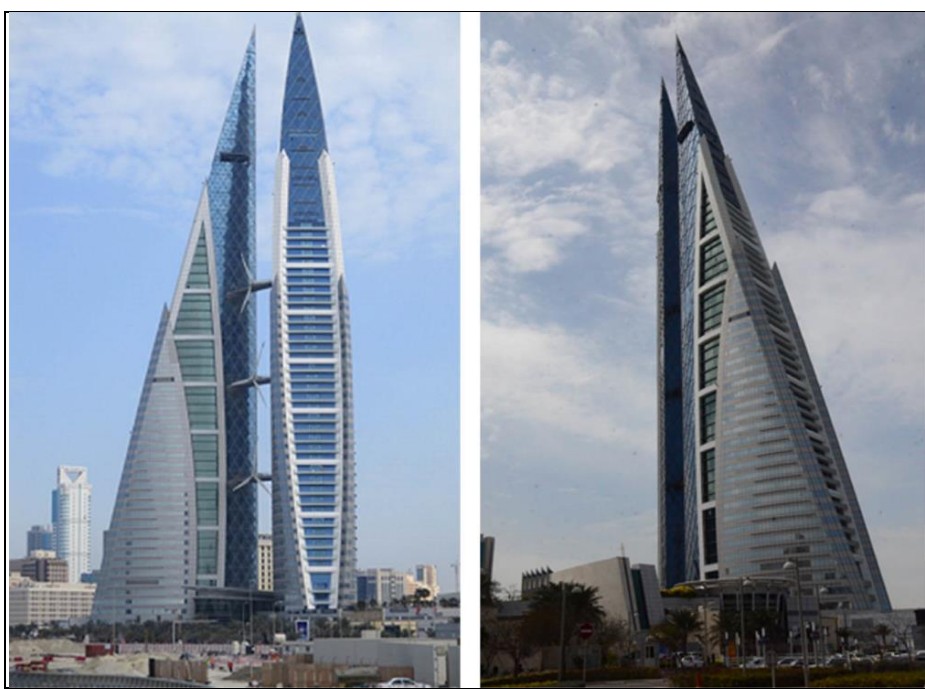

**Figure 24.** Bahrain World Trade Center (photographs by authors).

The elliptical form of the plan of towers and their profile cause the wind to act on them like the wings of an aircraft, creating a negative pressure that results in an increase in wind speed by up to 30%. This phenomenon has been effectively used in three wind turbines installed in buildings, which are oriented toward the extremely dominant prevailing wind. In conjunction with the shape of the towers and the velocity profile of the wind, the upper and lower turbines produce 109% and 93% of energy when compared to 100% for the middle turbine [53].

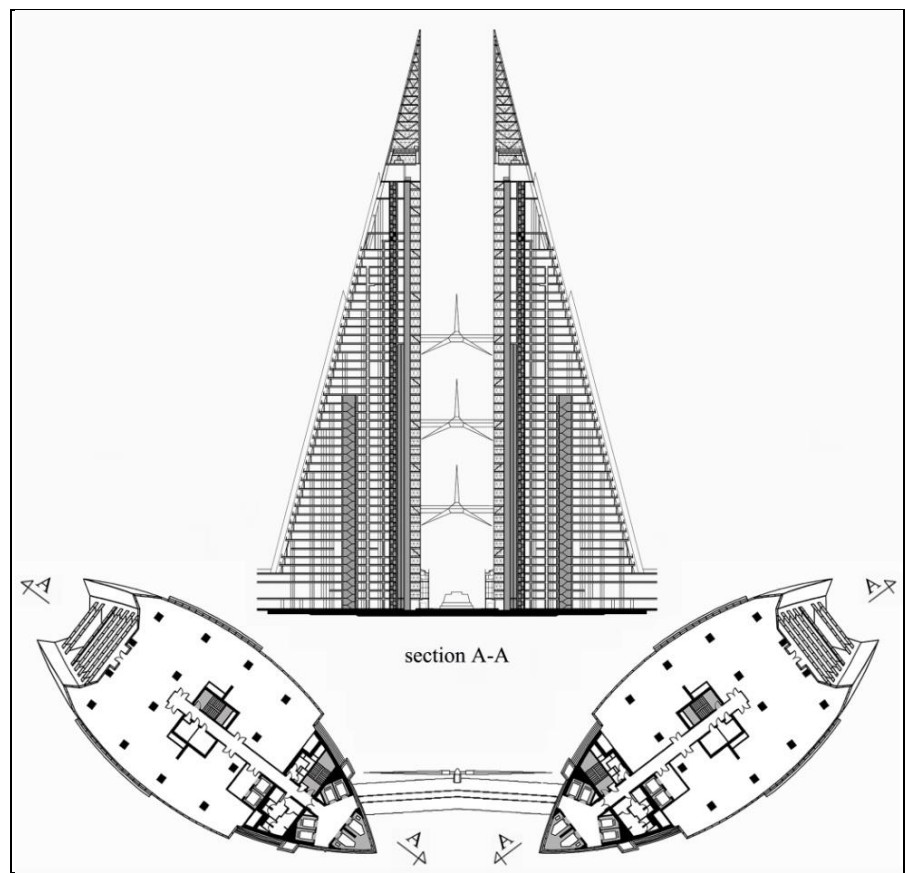

**Figure 25.** Bahrain World Trade Center: Floor plan and section (developed by authors on the basis of [54]).

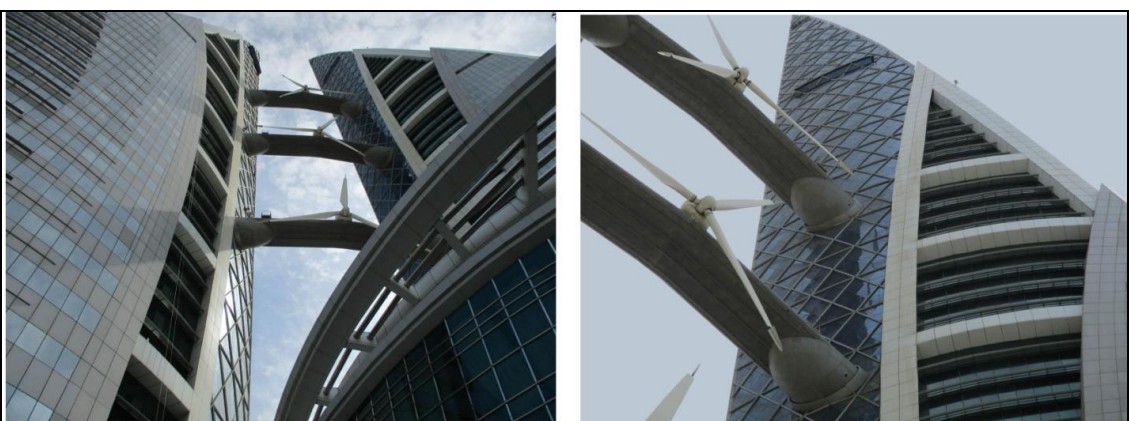

**Figure 26.** Bahrain World Trade Center: Three wind turbines supported on three different levels by bridges stretched between the towers (photographs by authors).

Besides wind energy, the Bahrain World Trade Centre building has other sustainable architecture elements [55]. The glass covering the building is high-quality solar glass with low shading to reduce the building's air temperature. The building is also connected to a district cooling system. The Bahrain World Trade Center takes seawater from the Persian Gulf, which is pumped through a pipeline to chilling units. The units then pass the chilled water through air conditioning units, which cool the air. There are also reflection pools at the entry points of the building, providing local evaporative cooling. Additionally, low-leakage, openable windows installed in the building support the mixed-mode operation in winter months. All these designs are cost-effective and reduce carbon emissions, as opposed to traditional heating and cooling systems.

### 4.6. Oasia Hotel Downtown (Singapore)—Advanced Green Facade System

Oasia Hotel Downtown is a mixed use high-rise building with reinforced concrete structure designed by studio WOHA Architects. The building is 193.3 m high and contains 27 floors at the above-ground level, Figure 27. Oasia Hotel Downtown has a cylindrical form, and is characterized by a new typology of a tropical high-rise with many terraces with gardens and vertical vegetation. Sky terraces on levels 6, 12, 21 and 27 offer ample public space for recreation social interactions throughout the tower [56].

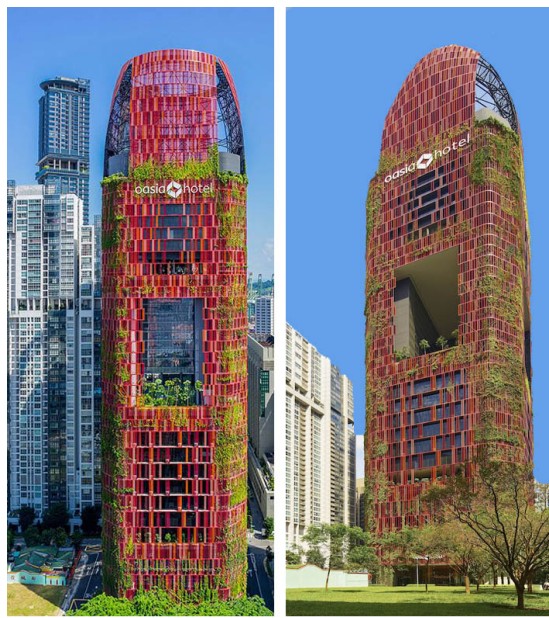

**Figure 27.** Oasia Hotel Downtown (photograph by authors).

The building has raft foundation that cooperates with piles. The load-bearing structure is a reinforced concrete slab-column, with four cores located in the corners of the truncated square plan, Figure 28. With the structural cores, which are located in the corners of the building, "sky terraces" enable a unique 360-degree view through the gardens to the city. This would not be possible with a typical centrally located core. By dividing the skyscraper into vertical segments, the sky terraces, together with the green facade, provide an ecological surface area of over 1000% in relation to the surrounding buildings. Sky terraces also serve as huge overhangs, directly shading the terrace below. The openness allows breezes to pass through the building for effective cross-ventilation.

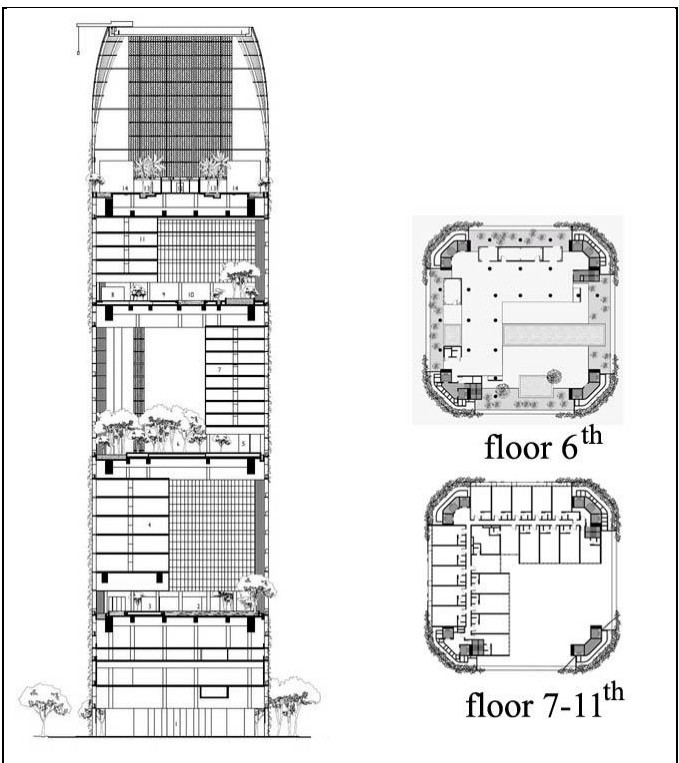

**Figure 28.** Oasia Hotel Downtown Floor plan and section (developed by authors on the basis of [56]).

The characteristic elements of the building are the L-shaped atriums appearing on every sky terrace that have a height of 21 to 35 m. Each atrium achieves an approximate ratio of height to depth of 1:1, providing a bright and airy environment during daylight. The external aluminum grid in five shades of red, which is attached to the facade of the building, allows the integration of various biological forms and the creation of a green cover (Figure 29). In addition, it also creates a contrast with the lush greenery and blue sky and enables the building to stand out among the numerous skyscrapers in the city center.

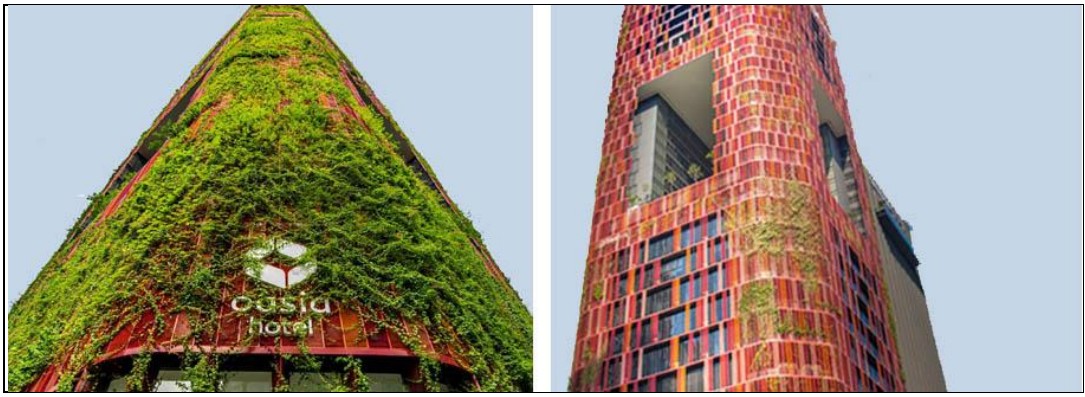

**Figure 29.** Oasia Hotel Downtown: Green facade (photograph by authors).

## 5. Discussion

Several dozen years ago, architects expressed their visions of an urban future based on new skyscraper typology. Le Corbusier was among them with his 1923 proposal of a series of 60-story office buildings and next for the Radiant City, which extended the concept more by specifying zones for working, living and rest. Le Corbusier's ideas for reshaping city centers have been for many years the foundation for high-rise public housing complexes in United States and Europe [57]. Another

tall building visionary was Frank Lloyd Wright, who proposed Illinois Tower with a high of 1 mile. While the idea was physically impossible then, Wright's vision became a touchstone in the ongoing race of skyscrapers height. The twenty first century has brought its own visions of the future [58]. The "Mile-Height" Kingdom Tower built near Jeddah (Saudi Arabia), reflects Wright's own scheme. The nearly mile-height Nakheel Tower proposed for Dubai replicates the features of the emirate's the highest building in the world Burj Khalifa. The Burj Mubarak al-Kabir in Kuwait is proposed to reach 1001 m, symbolizing "Arabian Nights" collection of stories.

The evolution of skyscrapers has become a trend that defines the nature of twenty first century cities. Most notable is the fact that skyscrapers are no longer an American phenomenon. Currently, high-rise buildings have been registered in 72 countries. Each city tries to build a skyscraper as an element of prestige and wealth. At the same time, it becomes an important landmark in the city and plays a major role in the technological development of modern architecture. After a tragic terrorist attack on the twin towers of the World Trade Center in New York, engineers have intensified their efforts to develop a super-safe construction. Technological progress has been clearly targeted, as exemplified by the safest building in the World.

The skyscraper is today in the most common form an Asian phenomenon. Tall buildings have spread well beyond Asia. Mirroring recent changes in the global economy, the Middle East has also adopted the new urban form. Hundreds of residential and mixed use high-rise buildings have been erected in Middle East over the last decade, many incorporating advanced designs and technology.

How high the next generation of skyscrapers will go is difficult to determine. Dubai's Burj Khalifa has 828 m of height, it is 60% taller than Taipei 101, the previous tallest building in the world. A number of supertall buildings will be completed in the next few years. Supertall buildings are extremely complicated to design, require a very robust leasing and sales market, and take more time to construct than most lenders can accept.

An important topic of discussion is how sustainable a high-rise tower can be. Although the trend to achieve net zero energy buildings, for which a balance between energy flow and renewable supply is established, the way to reach the goal is long. New skyscrapers in dense urban areas are generally greener than other types of commercial and residential buildings. They are typically located near mass transit, minimizing negative environmental impacts associated with road traffic. Vertical living also requires less energy for heat. Designers of skyscrapers continue to go to great lengths to minimize the environmental footprint of new buildings. These efforts take many forms: Orienting the building better to the sun and the wind, expanding the use of natural light and ventilation, providing thermal barriers in curtain wall design, maximizing the use of renewable energy (solar and wind), ensuring better collection and utilization of rainwater, and conserving energy through intelligent building managements systems.

The shape of today's skyscraper is particularly notable. The advances in technology and materials have allowed erection not only of very high buildings but also allowed them to take on new and exciting shapes. Today high-rise buildings can twist, lean and turn back on themselves. These shapes are chosen for visual effect, but occasionally they contribute to minimizing wind loads by improving a building's aerodynamic properties.

The skyscraper of the future will have a mixed-use function. The increasing popularity of mixed-use complexes, and in particular the growth in residential towers, has left its mark on every aspect of skyscraper design and construction. In terms of structure, concrete has now overtaken steel as the most prevalent skyscraper material. In terms of construction, mixed-use buildings are more difficult and costly to erect than single-purpose one. In terms of design, mixed-use buildings present the added complexity of segregating users and uses, taken into account pedestrian flows, vertical transportation, loading and other services. In designing these buildings, architects must often deal with multiple building code provisions, as standards for commercial and residential occupancy often differ.

Finally, it must be stated that knowledge about the advantages of high-rise buildings should remind us that although they are structures with advanced technology, they also cause shade that

prevents light from penetrating surrounding areas. Among other things, this problem is widely discussed by urban planners in New York and concerns the shading of Central Park by newly built skyscrapers. Therefore, a question arises regarding the future and direction of the development of high-rise buildings. However, one thing is certain, regardless of their future, they will always be a catalyst for technological development.

## 6. Conclusions

Technological innovations used in high-rise buildings can be manifested in many areas: Geometric form, construction, materials, vibration damping systems and energy efficiency. The development of computer technology has facilitated the design of high-rise buildings with complicated structural and functional solution forms. Increased computing power has allowed the creation of more advanced engineering programs, which for building models better simulate the actual behavior of a structure. This can especially be seen in high-rise buildings erected in the last years. Modern designs have broken the stereotypes of high-rise buildings in terms of history and tradition. An important aspect in the design of various architectural forms is the determination of the relationship between the shape of a building and the quality of its construction. Very often curvaceous shapes are inspired by various forms, which can occur in nature Capital Gate (Abu Dhabi, UAE) and Burj Khalifa (Dubai, UAE). A tall building, due to its shape, can be a very distinctive landmark in its environment and thus an easily recognizable building.

A simple measure of building quality is resistance to oscillating sway, resonance with the wind and also dynamic flexural stiffness. Complex shapes and requirements resulting from the height of buildings cause an increase in the load of constructional elements. Enhanced static and dynamic effects must be reflected in a properly selected construction system. At the end of the 19th century, the efficiency of diagonally braced elements that counteract lateral forces was taken into account when designing the first high-rise buildings. The use of the diagrid construction system is not new, but there is now a noticeable increase in the interest and application of this system in the design of tall buildings with large spans, especially concerning complex geometry. Diagrid structures do not require a core with high shear stiffness, because shear forces can be carried by the diagonal elements located on the perimeter of the structure. Perimeter diagrids carry horizontal and gravity loads and are used to support the edges of slab floors. This system is part of the trend of spectacular aesthetics, which can be exemplified by very iconic buildings (Hearst Tower, Capital Gate Tower, Doha Tower, The Bow, Swiss Re, etc.).

Undoubtedly, the least-resistant construction for an earthquake is a skyscraper, which is a certain paradox in comparison with their number in the world. The most modern skyscrapers in Tokyo are able to withstand earthquakes of over seven degrees on the Richter scale. Of course, more forces affect a building with a larger earthquake, and its construction therefore experiences larger displacements. A building's response to earthquakes is vibrations in the form of sinusoidal motion. In order to counteract both these forces and the impact of wind, apart from a rigid construction, very advanced technologies of damping devices are used. For example, the foundations of these buildings (Maison Hermes Tokyo) are mounted with a system of spring or elastomer vibration dampers, due to which tectonic movements affect the upper part of the building to a lesser extent. In addition, as presented by the characteristics of high-rise buildings, viscous oil dampers (Mode Gakuen Cocoon), anti-buckling steel stabilizers (Midtown Tower, Roppongi Hills, Kabukiza Tower) and tuned mass dampers (Tokyo Tree Tower) are used in various levels of these buildings. When using all these supporting elements, it is most important that the location of the center of gravity of the building does not change during earthquakes.

A very important aspect associated with the technological development of high-rise buildings is the safety of their users. One World Trade Center in New York is the most advanced building in the world when it comes to security technology, setting new standards for the design of high-rise buildings.

Sustainability is also a major issue concerning high-rise buildings [59]. It is strongly required to use sustainable concepts and applicable technology for reducing energy consumption and $CO_2$ emissions.

Covering the walls of a building with greenery affects the changing of microclimate, produces oxygen, absorbs $CO_2$ and captures particles of pollution [60,61]. Currently, plants are becoming an appropriate facade material in the creation of architecture. Their use is planned and dedicated to achieving both a specific aesthetic and ecological effect.

By the nature of high-rise buildings, it is very difficult to achieve a low energy building. High energy consumption in high-rise buildings has influenced the search for innovative solutions aimed at improving energy efficiency in this area. The research was focused on solutions based on renewable energy sources. Currently, photovoltaic panels and wind turbines are primarily used to produce electricity for a building's own needs. Designing the building together with an integrated wind turbine constituted a major design challenge for the Bahrain World Trade Center building. The project had to take into account the wind speed and direction, which occur in a given area, and as a result change parameters depending on the geometry of the building. There are many factors that affect the flow of wind in these installations. Among them are not only the location and occurring terrain, but also the shape of the building and its dimensions. Skyscrapers not only favor the development of innovative solutions, but also aim to improve human comfort when visiting a building or the safety of people residing in it. For example, the HMS (home management system) system is used, which integrates the majority of installations in an apartment. The organization of operation and modern equipment of high-rise buildings means that they belong to the category of "smart buildings".

**Author Contributions:** Both authors contributed the same to the analysis of the problem, discussion and writing the paper.

**Funding:** This research received no external funding.

**Conflicts of Interest:** The authors declare no conflict of interest.

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
