# Peer review of "Technological Advances and Trends in Modern High-Rise Buildings"

_buildings, doi:10.3390/buildings9090193_

Round 1

Reviewer 1 Report

The main objective of the article entitled "Review of Technological Advances and Trends in Modern High-Rise Buildings" is to provide a report of the current state of the technologies applied in skyscraper buildings, as well as to analyse the possibilities of innovative evolution from different fields: architectural , energetic, structural, constructive, etc.

This reviewer considers that the article does not reach its objective, being merely a descriptive document where randomly selected examples are provided in more than half of its content. Listed below are the reasons why this article should not be published in its current state, as well as observations made by the reviewer:

- Firstly, this document cannot be considered as a review. A review analyses a specific aspect of scientific knowledge, gathers the available bibliography and references in relation to the topic and updates the knowledge existing about it. However, this document does not analyse a specific aspect, but in a general and dispersed way it approaches technological advances. There is no sequence of references that collect the studies developed to date on this subject (most of the references refer to the examples of buildings presented in the article).

- In the introduction more references should be provided. There are certain documents that are mentioned and there is no reference (lines 34-37). Sometimes reference is made to “Leed” and sometimes to “LEED” àit is necessary to homogenize.

- The methodology simply refers to a collection of information that is not properly referenced and described.

- The whole section 3 referring to technological innovations would be much more interesting if each subsection were extended, adequately describing the systems described (images, diagrams, references ...). On page 3 there is a paragraph that almost occupies the entire page with too many loose ideas, without ordering and mixing information.

Sometimes it would help to use points or subsections to differentiate the ideas described. For example, section 3.3 lines 178-200 describes àPassive damping systems, which are divided into three categories. Afterwards, the description continues with other damping systems (tuned mass dampers, liquid dampers), so that everything is mixed and confusing.

- The selection of examples occupies 14 pages of the article. This section is limited to the description of each of the examples, without delving into anything. In addition, the examples appear to have been chosen randomly since they have no common characteristic. Its construction dates range from 2006 to 2016. 2006 could be considered an "old" building since it is 13 years old and much of the regulations applied at the time are now obsolete. The architectural typology is very different in each building, as well as the structural and constructive solution.

-The discussion section occupies only half a page. This section also includes ideas that should appear in “introduction” (line 700-702).

Other:

- pg 7 line 299: "insolation" incorrect -> insulation or isolation?

In summary, this reviewer considers that this article is not suitable for publication.

Reviewer 2 Report

The abstract should be reviewed to fully reflect what the paper is about. A recommendation is to remove the word 'comprehensive' as this is not achievable in one academic paper.

The introduction section reads well.

The methods section should be reviewed-  provide a narrative around the methodology chosen that led to the methods and provide a justification for their use. How were the types of modern high rise buildings chosen, self selecting?

Does the discussion add to the body of knowledge on this topic? More could have been offered in terms of the discussion.

The conclusion needs more work, what about some discussion that will contribute to the discourse of the future of high rise buildings?

Why introduce sustainability in the conclusion, some reference to it ealrier might have been appropriate.

Round 2

Reviewer 1 Report

This reviewer considers that the authors have managed to address all the comments and suggestions that were given in the review. The article has improved significantly, both in content and organization. The technical information added, and the subsections created are some of the changes that have contributed to the article successfully reaching its objectives.

Perhaps the final result is a bit extensive and some content could have been summarized (not only expanded), however, this reviewer considers this is a minor issue and the article is suitable for publication.

Reviewer 2 Report

The authors have made extensive revisions that make this paper a solid contribution now.